# Probing chromatin condensation dynamics in live cells using interferometric scattering correlation spectroscopy

Yi-Teng Hsiao[1], I-Hsin Liao [1,2], Bo-Kuan Wu[1], Hsueh-Ping Catherine Chu [2] & Chia-Lung Hsieh [1,3] ✉

Chromatin organization and dynamics play important roles in governing the regulation of nuclear processes of biological cells. However, due to the constant diffusive motion of chromatin, examining chromatin nanostructures in living cells has been challenging. In this study, we introduce interferometric scattering correlation spectroscopy (iSCORS) to spatially map nanoscopic chromatin configurations within unlabeled live cell nuclei. This label-free technique captures time-varying linear scattering signals generated by the motion of native chromatin on a millisecond timescale, allowing us to deduce chromatin condensation states. Using iSCORS imaging, we quantitatively examine chromatin dynamics over extended periods, revealing spontaneous fluctuations in chromatin condensation and heterogeneous compaction levels in interphase cells, independent of cell phases. Moreover, we observe changes in iSCORS signals of chromatin upon transcription inhibition, indicating that iSCORS can probe nanoscopic chromatin structures and dynamics associated with transcriptional activities. Our scattering-based optical microscopy, which does not require labeling, serves as a powerful tool for visualizing dynamic chromatin nano-arrangements in live cells. This advancement holds promise for studying chromatin remodeling in various crucial cellular processes, such as stem cell differentiation, mechanotransduction, and DNA repair.

A eukaryotic genome is stored in the cell nucleus as a DNA–protein complex called chromatin. The basic structural unit of chromatin is a nucleosome, a small bead with a diameter of 11 nm that consists of a segment of DNA wrapped around histone core proteins. The way that nucleosomes are assembled at the nanoscale directly impacts genome accessibility, gene expression, and DNA damage response[1,2]. Advanced microscopy techniques have deepened our understanding of the hierarchical structures of chromatin in cell nuclei. Electron microscope tomography with DNA labeling has shown that chromatin is a flexible and disordered chain of granules that ranges from 5 to 24 nm in diameter and can bend to achieve different levels of compaction[3,4]. Super-resolution fluorescence microscopy has disclosed various chromatin nanostructures in chemically fixed interphase nuclei[5–7]. State-of-the-art super-resolution fluorescence imaging of chromatin in live cell nuclei revealed distinct chromatin nanodomains with sizes ranging from 70 to 160 nm[8,9]. Although these high-resolution imaging methods reveal the chromatin ultrastructures at high spatial resolutions, they require extensive

sample manipulation of fluorophore labeling, and the achievable spatial resolution is restricted by the photon flux and motion blurring. The chromatin condensation levels in cell nuclei have also been inferred by analyzing the fluorescence confocal image of DNA stains (such as DAPI and Hoechst)[10]. However, DNA staining only allows for short-term observation or imaging of fixed cells because the DNA stains generally exhibit cellular toxicity by interfering with DNA replication and producing DNA damage[11].

Rather than resolving directly the chromatin nanostructures to determine their condensation levels, diffusion-based measurements infer nanoscopic chromatin configurations by analyzing the dependency of the chromatin mobility on chromatin compaction[12]. There are several well-established techniques available for probing the spatial diffusive motion of fluorescently labeled biomolecules in the live nucleus, such as fluorescence recovery after photobleaching (FRAP), fluorescence correlation spectroscopy (FCS)[13], and single-nucleosome tracking (SMT)[14,15]. Moreover, chromatin compaction states have been probed using Förster resonance

[1]Institute of Atomic and Molecular Sciences (IAMS), Academia Sinica, Taipei, Taiwan. [2]Institute of Molecular and Cellular Biology, National Taiwan University, Taipei, Taiwan. [3]Department of Physics, National Taiwan University, Taipei, Taiwan. ✉e-mail: clh@gate.sinica.edu.tw

energy transfer (FRET) and Fluorescence-lifetime imaging microscopy (FLIM)[16–18]. While FLIM and FRET are capable of detecting chromatin condensation that creates a change in FRET signal and fluorescence lifetime, their sensitivity to chromatin decompaction is relatively low because of the short working distances for FRET and FLIM signals. Finally, photobleaching and phototoxicity effects place a photon budget on the fluorescence-based measurements, limiting the total observation time, data acquisition rate, and data accuracy.

An alternative method to probe the chromatin organization and dynamics is by measuring its linear scattering light. Linear scattering signal is ubiquitous and stable, showing promise for label-free, long-term, and non-invasive imaging. Through interferometric detection, previous studies successfully measure temporal fluctuations of the scattering signals associated with the movements of intracellular organelles and biomacromolecules using dynamic full-field optical coherence tomography (FFOCT)[19–22] and partial wave spectroscopic microscopy (PWS)[23,24]. However, chromatin is weakly scattering and is enclosed by a nuclear membrane that generates a strong reflective background, making it challenging to detect chromatin-specific scattering signals. As a result, although these imaging methods are capable of measuring the scattering signals within cell nuclei, they generally lack the detection sensitivity and spatiotemporal resolutions for determining chromatin condensation at the nanoscale.

Our group recently demonstrated ultrasensitive chromatin imaging using an interferometric scattering (iSCAT) microscopy in transmission[25] referred to as coherent brightfield microscopy (COBRI)[26,27]. In COBRI, efficient sample illumination and signal detection is achieved by engineered laser scanning and point-spread function manipulation, leading to an enhanced optical signal for probing chromatin organization and dynamics. Using COBRI, we have shown that chromatin compaction at the nanoscale can be determined by analyzing the dynamic interference signal.[25] Unfortunately, as chromatin scattering recorded in the far field correlates to both the nucleosome density and its nanoscopic spatial arrangement, it has been nontrivial to measure unequivocally chromatin condensation in live cells based on the optical scattering signals.

In this work, we develop interferometric scattering correlation spectroscopy (iSCORS), an extension technique of iSCAT microscopy, that enables the quantitative assessment of nanoscale chromatin compaction states by analyzing dynamic light scattering (DLS) signals generated from unlabeled live mammalian cell nuclei. Inspired by the early works on DLS microscopy[28,29], iSCORS employs a correlation spectroscopic analysis to measure the diffusion coefficient and density of chromatin, which facilitates quantitative determination of chromatin condensation states. Additionally, we have upgraded our earlier system by incorporating a high numerical aperture microscope condenser for laser illumination. This enhancement has allowed us to achieve three-dimensional (3D), label-free imaging of chromatin, significantly improving spatial resolution. Utilizing iSCORS, we measure chromatin condensation dynamics in response to transcription inhibition, characterizing the influence of gene transcription on nanoscale chromatin structures. Furthermore, leveraging the inherent advantages of label-free and non-invasive iSCORS imaging, we examine the cell heterogeneity in chromatin condensation levels. Finally, we explore the temporal variations in chromatin condensation of individual cells over an extended period, unveiling considerable spontaneous fluctuations. As a whole, this study showcases iSCORS microscopy as a powerful tool for investigating the dynamic reorganization of chromatin in living cells.

## Results
### Interferometric scattering correlation spectroscopy (iSCORS) probes nanoscopic chromatin fluctuation in living cells without labeling
The optical setup of iSCORS is a custom-built COBRI microscope, an iSCAT microscope in transmission[26,27], integrated with a simultaneous epifluorescence imaging channel (Fig. 1a and "Methods"). In this microscope technique, the specimen is illuminated with a laser at a visible wavelength, and the transmission image is captured by a camera at a high

speed of 5000 frames per second (fps) unless stated otherwise. The transmission geometry avoids the complications introduced by the reflective background generated by the nuclear membrane[30]. To detect the signal of weakly scattering chromatin, we enhance the interference contrast by back-pupil function engineering ("Methods")[27]. Moreover, the coherence noise of laser illumination is minimized through fast scanning of a focused laser beam with acousto-optic deflectors. This approach effectively reduces the spatial coherence of illumination, leading to an enhanced spatial resolution in nuclear imaging (Supplementary Fig. 1). In this study, we used a condenser lens with a numerical aperture (NA) of 0.8 and an imaging objective with an NA of 1.49, resulting in a high spatial resolution of ~240 nm laterally and ~1 μm axially (Supplementary Fig. 2).

The transmission image of a cell records the combined intensities of the forward scattered light of the sample (the signal, $|E_s|^2$) and the non-scattered transmitted beam (the reference, $|E_r|^2$), along with their interference ($2|E_r||E_s|\cos\theta$):

$$I_{\text{det}} = |E_r + E_s|^2 = |E_r|^2 + |E_s|^2 + 2|E_r||E_s|\cos\theta, \quad (1)$$

Here, $\theta$ is the phase difference between the signal field and the reference field. As chromatin is weakly scattering, $|E_s|^2$ is negligible in the detected intensity. To quantify the interference contrast, we calculate an interference contrast (C) by normalizing the measured interference intensity with the reference beam intensity:

$$C = \frac{I_{\text{det}} - |E_r|^2}{|E_r|^2} \cong 2\frac{|E_s|}{|E_r|}\cos\theta. \quad (2)$$

The reference beam is spatially uniform with a slow intensity variation due to inhomogeneous illumination and can be effectively estimated by a flat field correction (Supplementary Fig. 3). Figure 1b plots the C map of a live human bone osteosarcoma epithelial cell (U2OS cell). The cell nucleus is prominent in the C map because the nuclear membrane provides a strong scattering signal. Meanwhile, the nuclear bodies of nucleoli within the nucleus are also clearly visible because of their distinct molecular densities (indicated by white arrows in Fig. 1b).

In addition to the large nuclear structures that can be directly visualized in the C map, iSCORS microscopy measures a DLS signal as a temporal fluctuation of C in the nucleus (on the order of 0.01) caused by the diffusive motion of biomolecules including chromatin. To quantify the DLS signal (denoted as $C_{\text{DLS}}$), we normalized C by its temporal average and then minus one:

$$C_{\text{DLS}} = C/\langle C \rangle_t - 1. \quad (3)$$

where $\langle C \rangle_t$ denotes the temporal average of C over an observation time of 1 s. This choice of observation time of 1 s is a compromise to ensure sufficient statistical data is gathered for calculating the time average while excluding scattering signal changes that arise from cell movements and are not relevant to chromatin dynamics. Figure 1c displays the DLS signal of the cell nucleus that appears as time-varying random interference patterns because the chromatin is densely packed with structures that are much smaller than the optical diffraction limit of visible light.

We extract two temporal features from the DLS signal, the correlation time and the fluctuation magnitude, which are useful for probing nanoscopic chromatin architectures. The correlation time is directly connected to the diffusion coefficient of chromatin. Early studies show that chromatin in mammalian cells undergoes anomalous subdiffusion, measured by tracking single nucleosome and specific genomic loci[14,15,31,32]. In this study, we choose to measure an apparent diffusion coefficient, denoted as $D^*$, at the shortest timescale of iSCORS measurement because of its high sensitivity to the nanoscopic chromatin configuration. In the timescale of our iSCORS imaging at milliseconds, chromatin movement is on the nanometer scale that corresponds to local diffusion driven by thermal fluctuation. Thus, we

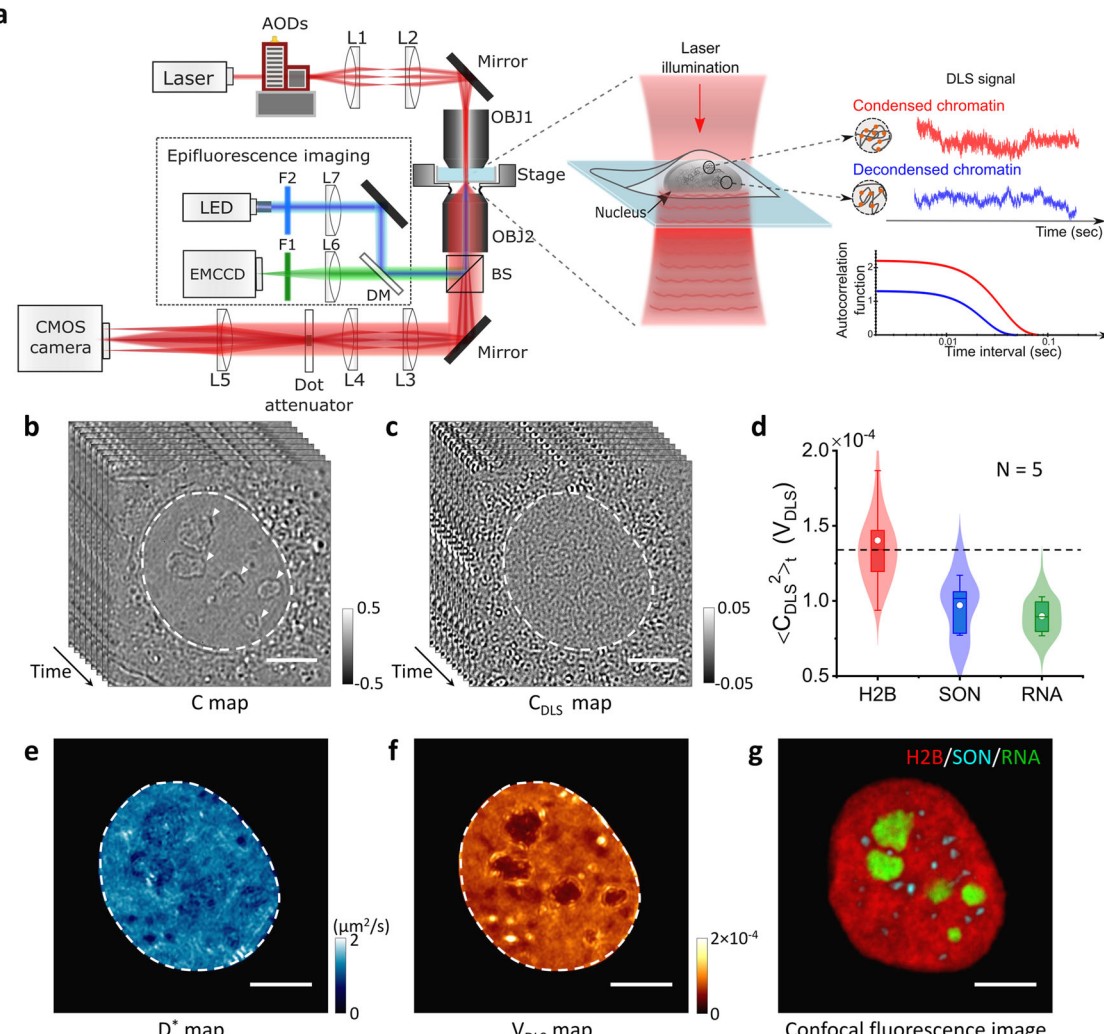

**Fig. 1 | Label-free iSCORS imaging resolves chromatin density and dynamics in a single live cell nucleus. a** Schematic diagram of iSCORS microscopy. Focused laser light is rapidly scanned across the cell sample by two-axis acousto-optical deflectors (AODs) through a condenser objective lens (OBJ1). A transmission image containing dynamic scattering signals is collected by an objective lens (OBJ2) and projected onto a high-speed CMOS camera. Chromatin dynamics and condensation levels are deduced based on the dynamic signals through correlation spectroscopy analysis. To enhance the interference contrast of chromatin, a dot-shaped attenuator is inserted in the Fourier plane of the sample, selectively reducing the intensity of the non-scattered transmitted light. Fluorescence imaging is integrated through a beamsplitter (BS) for simultaneous epifluorescence observation. Normalized interference contrast map (C map) (**b**) and DLS image ($C_{DLS}$ map) (**c**) of a cell nucleus. **d** Comparison of the temporal variance of $C_{DLS}$ generated by chromatin (H2B), nuclear speckles (SON), and nucleoli (RNA). The distributions are plotted using kernel smoothing methods. Calculated $D^*$ map (**e**), $V_{DLS}$ map (**f**), and a confocal fluorescence image (**g**) of chromatin (H2B, red), nuclear speckles (SON, cyan), and nucleoli (RNA, green). Note that the confocal fluorescence images are acquired by a separate fluorescence confocal microscope (Yokogawa CSU) using a ×100 objective with NA 1.45. The nuclear regions, indicated by white dashed circles in (**b, c, e, f**), are identified in the C map assisted by a machine learning algorithm ("Methods"). The $D^*$ and $V_{DLS}$ maps are filled with zeros in the areas external to the nuclear boundaries. The nucleoli are marked by white arrows in (**b**). Scale bars are 5 μm.

reason that, by analyzing the iSCORS signal with DLS models, the $D^*$ of chromatin at the nanoscale can be determined. Specifically, our iSCORS imaging resembles a DLS measurement with a coherent reference beam[33]. Based on the DLS modeling, we estimate $D^*$ by calculating the temporal autocorrelation function (ACF) of the contrast fluctuation of every pixel.

$$ACF_{DLS}(\tau) = \frac{\langle C_{DLS}(t)C_{DLS}(t+\tau)\rangle_t}{\langle C_{DLS}(t)^2\rangle_t} \qquad (4)$$

By calibrating the system with nanoparticle colloids of known diffusion coefficients, we are able to measure the $D^*$ of chromatin based on the ACF (see derivation in Supplementary Note 1 and nanoparticle data in Supplementary Fig. 4).

$$ACF_{DLS}(\tau) = e^{-D^*k^2\tau} \qquad (5)$$

Here $k$ denotes the wavenumber of the scattering detection that is determined by the optical setup and illumination wavelength.

Figure 1e displays the $D^*$ map of a cell, exhibiting spatial variation in nanoscopic chromatin mobility in the nucleus. An apparent diffusion coefficient of $0.92 \pm 0.07$ μm²/s is measured for chromatin in the millisecond timescale. Such $D^*$ is approximately an order of magnitude slower than the diffusion of an inert protein in the cell nucleus, suggesting that iSCORS measures the diffusion of chromatin fibers instead of freely diffusing nuclear proteins[34]. Furthermore, iSCORS measurement is in a millisecond timescale, capturing local molecular diffusion events that are faster than the chromatin movements measured by fluorescence-based methods, such as tracking and bleaching experiments that typically work over a much longer timescale of tens to hundreds of milliseconds[35]. This is due to the subdiffusive motion of chromatin, leading to a decreasing $D^*$ when the timescale increases[14,15,36]. We confirm this phenomenon by increasing the measurement timescale of iSCORS to 5 ms, and indeed a significantly

reduced $D^*$ is measured, consistent with the behavior of chromatin sub-diffusion (Supplementary Fig. 5).

Another feature we extract from the DLS signal is the fluctuation magnitude which scales with the time-varying scattering field. We quantify the fluctuation magnitude of the DLS signal by calculating its temporal variance, denoted as $V_{DLS}$. For a monodisperse nanoparticle colloid, $V_{DLS}$ scales linearly with the particle mass density[25]. In analogy to nanoparticle colloids, $V_{DLS}$ of a live cell nucleus is expected to measure the chromatin density. With the guidance of fluorescence labels, we confirm a high correlation between the $V_{DLS}$ map and the confocal fluorescence chromatin image (H2B-mCherry), both of which resolve numerous chromatin-depleted regions that correspond to nucleoli and RNA-filled compartments (Fig. 1f, g, more image data in Supplementary Fig. 6)[37]. Meanwhile, we measure much weaker DLS signals from RNA-protein complexes, such as nucleoli (labeled by SYTO RNASelect) and nuclear speckles (immunostained against the nuclear protein SON), compared to those from chromatin (Fig. 1f, g, more in Supplementary Fig. 6). Figure 1d displays the quantitative comparisons of the $V_{DLS}$ signals created by chromatin, nuclear speckles, and nucleoli, showing that chromatin contributes the strongest signal. We believe this is because protein–RNA complexes lack high-ordered structures like chromatin, and thus the measured DLS signal is dominated by the large and dense molecular complexes of chromatin. It is worth noting that the fluorescence image of H2B does not resolve nanoscopic chromatin configuration and condensation states due to the limited spatial resolution. On the other hand, $D^*$ and $V_{DLS}$ maps capture chromatin fluctuations at the nanoscale, enabling us to determine chromatin condensation levels (as detailed in the next section).

## iSCORS imaging quantitatively measures chromatin condensation levels in live cell nuclei

We introduce a robust characterization of chromatin condensation level using $D^*$ and $V_{DLS}$ calculated from a single iSCORS measurement. In the case of chromatin condensation where nucleosomes are tightly packed together, the scattered signals of individual nucleosomes add up coherently, leading to an enhanced DLS signal of a greater $V_{DLS}$. Quantitatively, $V_{DLS}$ scales with the volume of chromatin nanocluster ($V_{DLS} \propto a_c^3$, $a_c$ is the radius of the chromatin nanocluster) when the overall mass density remains constant[25]. Meanwhile, chromatin condensation also enlarges the nanocluster size, reducing the chromatin mobility $D^*$ that scales with $1/a_c$, assuming thermally driven free diffusion of chromatin at the nanoscale. Therefore, a change in chromatin condensation level will be measured as variations in $V_{DLS}$ and $D^*$ following the dependency of $V_{DLS} \propto (1/D^*)^3$.

To validate our method, we perform iSCORS imaging on single U2OS cell nuclei and measure their $D^*$ and $V_{DLS}$. For quantitative analysis, we calculate the spatial median values within the nuclear region to represent the global chromatin configuration, mitigating bias from nonspecific and spatially localized nuclear structures such as nucleoli and other RNA-protein complexes. In addition to cells under normal conditions, we induce global chromatin decondensation by inhibiting histone deacetylase activities with sodium butyrate (NaB) and Trichostatin-A (TSA) ("Methods"). On the other hand, global chromatin condensation is induced by treating cells with Actinomycin D (ActD), and sodium azide and 2-deoxyglucose (2-DG +NaN₃) ("Methods"). The effectiveness of these treatments on modifying chromatin condensation states is confirmed by analyzing the fluorescence intensity distribution of DNA stains based on coefficient of variation (CV), a method previously established to estimate chromatin compaction (Method and Supplementary Fig. 7)[38]. Representative cell images of different treatments are displayed in Fig. 2a. Compared to the untreated cells, we measure a decreased $1/D^*$ together with a lower $V_{DLS}$ for decondensed chromatin, consistent with a smaller chromatin nanocluster of loosely packed chromatin. In contrast, an opposite transformation is observed when inducing chromatin condensation by treating cells with ActD and 2-DG+NaN₃. We note that chromatin condensation changes are difficult to detect in the confocal fluorescence H2B image due to the limited spatial resolution.

We display the $V_{DLS}$ and $1/D^*$ for single nuclei under different treatments in a log-log $V_{DLS}$-$1/D^*$ plot (Fig. 2b), exhibiting a positive correlation between $V_{DLS}$ and $1/D^*$. A close examination of the data reveals that the data points fall on a line with a slope of three, corresponding to a scaling of $V_{DLS} \propto (1/D^*)^3$. Such a dependency agrees well with our model and with the nanoparticle colloids data (Supplementary Fig. 8), indicating that the chemical treatments alter the chromatin condensation states at the nanoscale and these changes can be detected by iSCORS imaging. We quantitatively determine the chromatin condensation level by projecting the data points onto a line with a slope of three in the log–log plot of $V_{DLS}$-$1/D^*$ (inset of Fig. 2b, see "Methods" for the details of quantification). Using this quantification, we reliably measure the chromatin condensation change induced by chemical drug treatments (Fig. 2c). We stress that although both $V_{DLS}$ and $D^*$ are sensitive to chromatin condensation states, they may be individually influenced by other chromatin properties, such as mass density and chromatin rigidity, that are indirectly related to chromatin compaction. Consequently, our method that combines $V_{DLS}$ and $1/D^*$ helps enhance the accuracy of chromatin condensation measurements. For the rest of our investigations, we will utilize the aforementioned method to assess and characterize chromatin condensation levels.

It is informative to examine the meaning of the $y$-intersect value of the line with a slope of three in the log-log plot of $V_{DLS}$ against $1/D^*$. From the iSCORS model and nanoparticle data (Supplementary Fig. 8), the $y$-intersect value indicates the overall mass density. We validate the interpretation of the $y$-intersect value as mass density by mapping the $y$-intersect value of a cell nucleus onto a spatial representation. The derived $y$-intersect map shows a high correlation with the chromatin fluorescence image, supporting our model that the $y$-intersect denotes mass density (Supplementary Fig. 9). The chromatin mass density of individual cells is consistent, within a variation of ~15%, set by the inherent cell heterogeneity and measurement uncertainty (Supplementary Fig. 10). Interestingly, the mass densities of different mammalian cell lines also exhibit a similar constancy, with a variation of less ~10% (Supplementary Fig. 11).

## Optical sectioning in iSCORS enables 3D mapping of chromatin condensation

A distinctive feature of iSCORS imaging is its ability to achieve a high axial resolution in the reconstructed chromatin condensation maps, even though iSCORS microscopy utilizes a transmission widefield configuration. This phenomenon arises because the DLS signal is most sensitive to the molecular fluctuations at the focal plane where the Gouy phase shift is significant, serving as a mechanism for rejecting the out-of-focus signals[39]. We experimentally assess the optical sectioning capability of the condensation map by acquiring $z$-stack images of a cell nucleus (Fig. 3a–c). The axial resolution is determined based on the line spread function at the cell–coverglass interface, resulting in an axial resolution of ~1.1 μm (Supplementary Fig. 12). Notably, this resolution is comparable to that of a fluorescence confocal microscope (Yokogawa CSU-X1) employing a similar numerical aperture of the optical microscope objective (Fig. 3d–f). The 3D imaging capability of iSCORS is illustrated in a supplementary video (Supplementary Movie 1).

## iSCORS detects chromatin condensation dynamics associated with gene transcription activities

We take a step further by investigating the impact of gene transcriptional activities on chromatin architecture. Prior research has established that chromatin undergoes remodeling to facilitate access to genomic DNA by transcription factors and RNA polymerases[40,41]. In this study, we explore alterations in chromatin condensation resulting from the introduction of chemical compounds that suppress gene transcription. Specifically, we employ 5,6-dichloro-1-beta-D-ribofuranosylbenzimidazole (DRB)[42], α-Amanitin (α-AM)[43], and ActD[44] to inhibit gene transcription ("Methods"), which block RNA polymerase II (RNA pol II) transcription through different mechanisms[45–47].

**Fig. 2 | Quantitative detection of chromatin condensation levels of single live cell nuclei by iSCORS imaging. a** Representative $V_{DLS}$, $D^*$, and chromatin condensation maps along with confocal fluorescence images of chromatin (H2B-mCherry) in cell nuclei under different chemical treatments that modify the global chromatin condensation states. The nuclear region is segmented for better visualization. Scale bars are 5 μm. **b** $V_{DLS}$-1/$D^*$ scatter plot of single live cell nuclei under different chemical treatments. Each data point represents the spatial median value of $V_{DLS}$ and 1/$D^*$ within the region of a cell nucleus. The data points fall approximately on a line with a slope of three, indicating that the chromatin condensation states are modified by the chemical treatments but the overall chromatin density remains unchanged. Chromatin condensation level is determined based on $V_{DLS}$ and 1/$D^*$ by projecting the data points onto a line with a slope of three (see the inset and "Methods"). **c** Box plot of chromatin condensation levels in single-cell nuclei under treatments that modify the global chromatin condensation, showing statistically different results from the untreated cells.

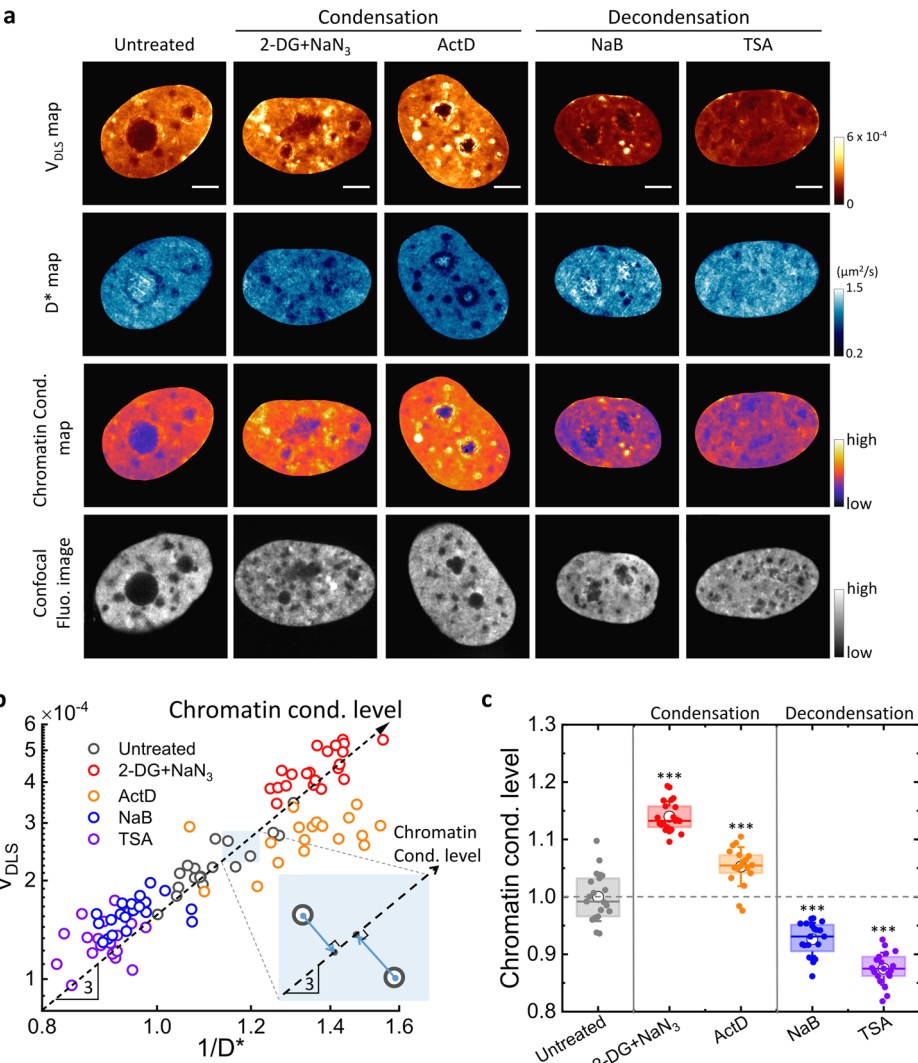

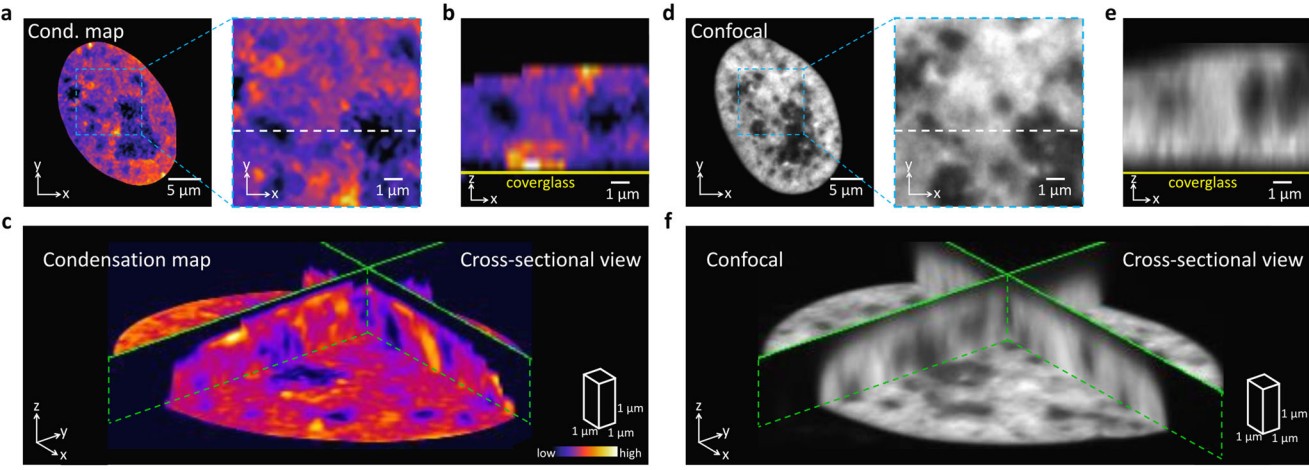

**Fig. 3 | Optically sectioning of iSCORS microscopy enables 3D visualization of chromatin organization. a** Chromatin condensation map (*xy* plane) of a cell nucleus, with close-up revealing chromatin-depleted regions. **b** Cross-sectional view (*xz* plane) of the chromatin-depleted regions along the dashed line shown in (**a**),

illustrating their 3D structures. **c** 3D cross-sectional view of the reconstructed chromatin condensation map of a cell nucleus. Fluorescence confocal images of H2B-mCherry of the same nucleus in the *xy* plane (**d**), *xz* plane (**e**), and 3D cross-sectional view (**f**), validating similar patterns in chromatin structures.

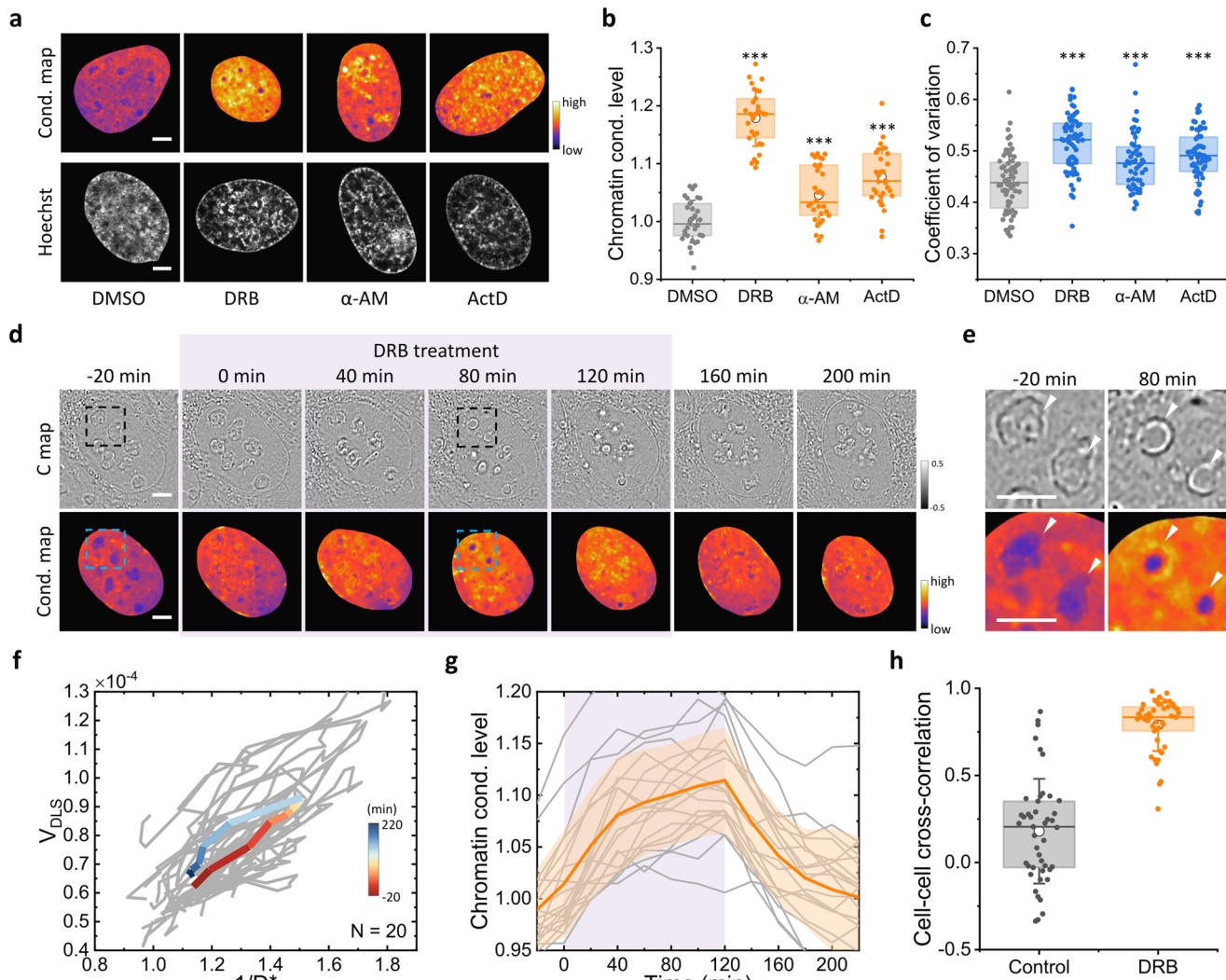

**Fig. 4 | Dynamic chromatin remodeling induced by transcription inhibition.**
**a** Chromatin condensation maps and Hoechst confocal fluorescence images of representative cell nuclei under different treatments that inhibit transcription (DRB, α-AM, and ActD) and the control (DMSO). **b** Chromatin condensation levels for different transcription inhibitors measured by iSCORS. **c** Coefficient of variation (CV) for different transcription inhibitors calculated based on the Hoechst fluorescence images. **d** Time-lapse iSCORS images of a representative cell nucleus continuously monitored during the addition and removal of DRB. The DRB is added at 0 min and removed at 120 min. The dashed squares mark the regions for close-up views displayed in (**e**). **e** Close-up views of a subnuclear region containing two nucleoli, indicated by white arrows, before and after the addition of DRB.

**f** Individual trajectories (gray) and averaged trajectory (colored) of $V_{DLS}$ and $1/D^*$ of 20 cells, showing chromatin condensation upon DRB addition and chromatin decompaction after DRB removal. **g** Time traces of chromatin condensation level. Gray curves represent the data from individual cells, while the orange curve and shaded areas mark their averages and standard deviations, respectively. The background color for 0–120 min indicates the period of DRB treatment. **h** Cell–cell cross-correlation of chromatin condensation levels. DRB-treated cells exhibit synchronized chromatin condensation dynamics due to modifications of transcription activities, whereas untreated cells show no cross-correlations between cells. Scale bars in (**a**, **d**, **e**) are 5 μm.

Utilizing iSCORS, we measured enhanced chromatin condensation levels upon the introduction of transcription inhibitors to U2OS cells (Fig. 4a, b). These observations substantiate the prevailing concept that gene transcription involves the unwinding of chromatin packaging to expose the DNA, and thus inhibiting transcription leads to a more densely compacted and tightly closed chromatin nanostructure. Our observation of chromatin compaction upon blocking the transcription activity of RNA Pol II is consistent with previous works, revealing a close connection between chromatin organization and transcription activities. For example, active RNA polymerase is found essential for producing and maintaining decondensed chromatin[48]. Additionally, chromatin compaction induced by transcription inhibitors has been measured by fluorescence-based techniques[49,50]. The enhanced chromatin condensation in our experiments is further supported based on an analysis of fluorescence intensity distribution of DNA stain of the cell nuclei (Fig. 4c and Supplementary

Fig. 13)[38]. Compared to these fluorescence-based methods, the significance of iSCORS imaging lies in its label-free nature, thus facilitating the observation of chromatin remodeling within native live cells, devoid of the potential perturbations that fluorophore labeling may introduce.

Furthermore, we demonstrate that label-free and noninvasive iSCORS imaging enables the real-time monitoring of chromatin condensation dynamics during the addition and subsequent removal of the transcription inhibitor, DRB. While previous studies measure chromatin condensation change induced by DRB by using fluorescence lifetime-based approaches[49], acquiring chromatin condensation dynamics has been difficult due to the effects of photobleaching and phototoxicity. We use iSCORS to measure the chromatin condensation levels at 20-min intervals, at an image acquisition rate of 1000 fps, over a period spanning up to 4 h. After a 120-min incubation period with DRB, we remove the inhibitor by exchanging the buffer medium, allowing the cells to return to their states before the treatment. Continuous

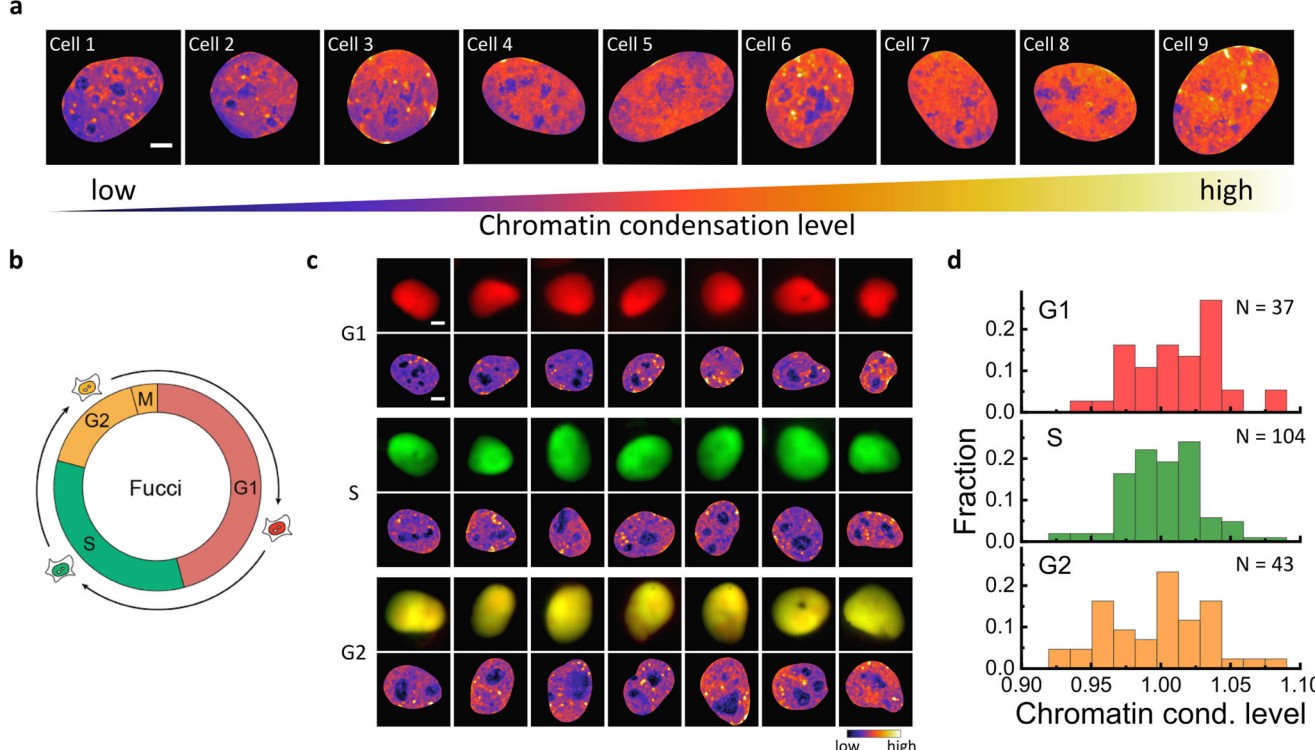

**Fig. 5 | Cellular heterogeneity in chromatin condensation levels among interphase cells. a** Chromatin condensation maps of representative U2OS cells that exhibit significant variations in condensation levels. **b** Schematic diagram of cell cycle phases with color representations of G1 (red), S (green), and G2 (orange) phases. **c** Representative fluorescence images and chromatin condensation maps of cell nuclei in the G1, S, and G2 phases. Scale bars are 5 μm. **d** Chromatin condensation levels of single nuclei measured in G1, S, and G2 phases. The differences between the three phases are statistically insignificant ($P > 0.05$).

time-lapse iSCORS imaging captures the entire process of chromatin remodeling in individual nuclei induced by transcription inhibition and subsequent recovery. As depicted in Fig. 4d, time-lapse iSCORS image data from a representative cell nucleus subjected to transcription inhibition reveals persistent chromatin condensation upon the addition of DRB. Following the 120-min interval, the removal of DRB initiates a chromatin decompaction process that coincides with the restoration of transcriptional activities. This results in the chromatin condensation level returning to its initial state by the 200-minute mark. This recovery serves as additional evidence that the observed chromatin condensation is not a result of toxicity.

The high spatial resolution of the chromatin condensation map reveals changes in the morphology of nucleoli following DRB treatment. Specifically, nucleoli become smaller and adopt a more spherical shape, as illustrated in Fig. 4e. This finding aligns with previous studies that have shown how transcription inhibition can alter nucleolar structures by modifying their liquid condensate phases[51].

Figure 4f presents the temporal profiles of $V_{DLS}$ and $1/D^*$ for 20 cells, all of which exhibit analogous responses to DRB treatment. Consequently, this leads to a strong cell-cell cross-correlation in chromatin condensation dynamics, measured at $0.79 \pm 0.15$, in contrast to the negligible correlation observed in untreated cells (Fig. 4h). The observed chromatin condensation dynamics induced by DRB treatment are depicted in Fig. 4g, providing clear evidence of chromatin condensation as a direct consequence of transcription inhibition by DRB. We also confirm that such changes in chromatin condensation do not manifest in a control experiment involving the solvent of DMSO (Supplementary Fig. 14). Finally, we note that the DRB treatment slightly reduces the degree of cellular heterogeneity in chromatin organization and dynamics, decreasing it from ~7.5% to ~4.5%. Taken together, our data show that gene transcription activities significantly influence chromatin condensation dynamics and contribute, at least partially, to the cellular heterogeneity in chromatin condensation states.

## Interphase cells exhibit heterogeneous chromatin condensation levels

In addition to monitoring the dynamics of chromatin condensation, iSCORS imaging provides an opportunity for quantitative comparison of chromatin condensation levels in individual native cell nuclei, circumventing complications introduced by fluorophore labeling and other sample manipulations. Our iSCORS dataset reveals substantial variations in global chromatin condensation levels within the U2OS interphase cells. Figure 5a illustrates chromatin condensation maps of representative cell nuclei, highlighting the pronounced cellular heterogeneity in chromatin condensation levels within interphase cells without any treatments. The diversity of chromatin condensation states observed across different cells through iSCORS imaging aligns with previous studies that investigated various chromatin states in individual cells based on single-cell histone modification patterns[52–54]. These techniques provide opportunities to reveal cellular states under various physiological or pathological conditions, allowing for the detection of unknown or rare cell types, and unraveling cell-type-specific differences and dynamics. Moreover, heterogeneity in chromatin compaction, observed in stem cells via fluorescence microscopy, has been found to reflect their differentiation status[55]. In our study, we examine chromatin compaction in a stable U2OS cell line, which is considered a homogeneous cell sample with minimal cellular heterogeneity. The considerable differences in chromatin condensation levels among individual cells, as revealed in our iSCORS data, are unexpected. This underscores the inherent cellular heterogeneity in chromatin organization, a finding that has not been available until now.

To determine whether the heterogeneous chromatin configuration is associated with distinct cell cycle stages, we access chromatin condensation levels in single nuclei throughout various cell phases using HeLa cells marked with the fluorescent ubiquitination-based cell cycle indicator (Fucci)[56]. With Fucci, we can identify the cell phase (G1, S, and G2 phases) of a single nucleus based on the ratios of red and green fluorescence intensities (as depicted in Fig. 5b), while simultaneously conducting iSCORS imaging of chromatin.

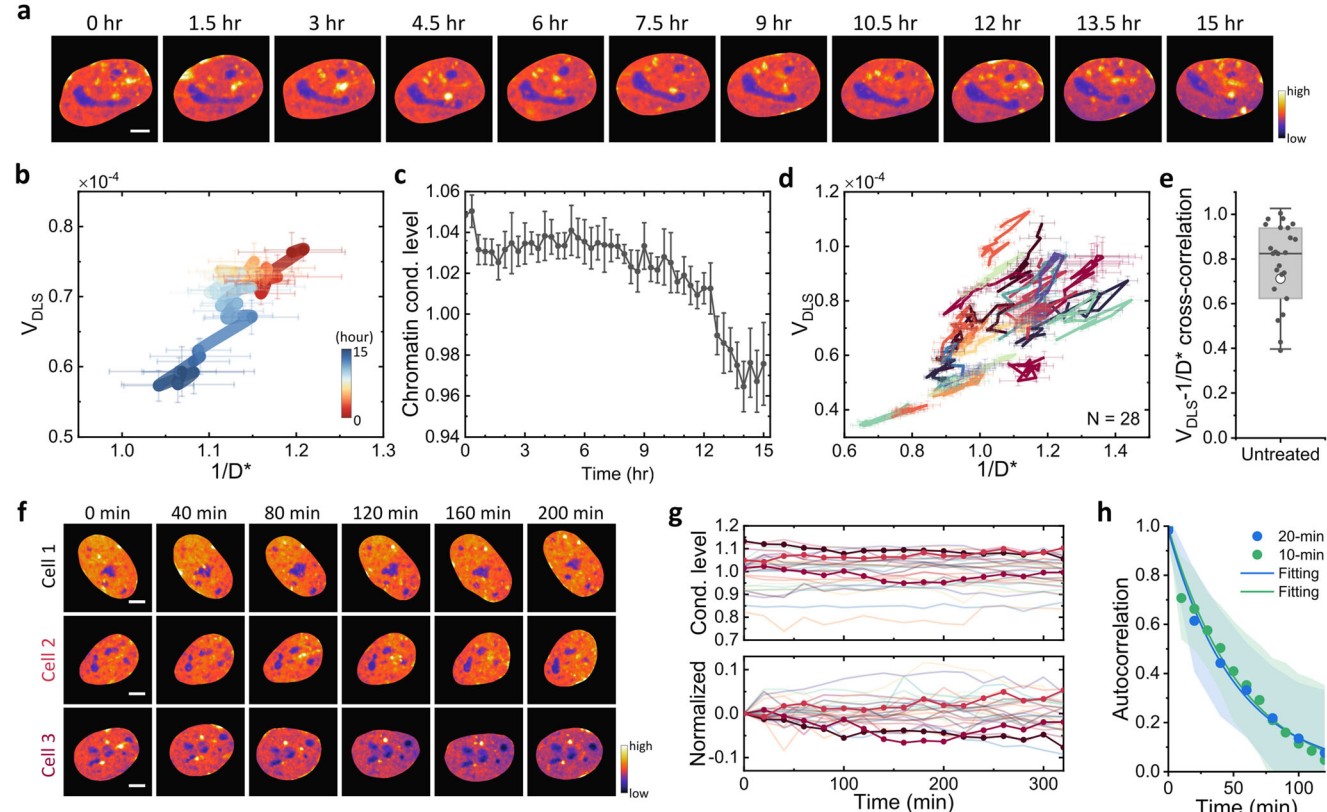

**Fig. 6 | Long-term iSCORS imaging reveals dynamic chromatin condensation levels in single live cell nuclei. a** Time-lapse chromatin condensation maps of a representative U2OS cell nucleus over a 15-h period. Scale bar: 5 μm. **b** Trajectories of $V_{DLS}$ and $1/D^*$ reconstructed from 15-hour iSCORS image data for the representative nucleus. **c** Chromatin condensation dynamics of the cell deduced from iSCORS imaging. Error bars represent the measurement uncertainty (standard deviation) of iSCORS imaging. **d** Time-lapse iSCORS data of 28 U2OS cells depicted in a $V_{DLS}$-$1/D^*$ plot. **e** Cross-correlation between $V_{DLS}$ and $1/D^*$ for 28 cells, demonstrating a strong positive correlation indicating dynamic changes in chromatin condensation levels. **f** Time-lapse chromatin condensation maps of three representative cells, exhibiting distinct condensation dynamics. **g** Time traces of chromatin condensation dynamics of 28 cells (top) and their normalized fluctuations (bottom). The three colored traces highlight the data of the three cells depicted in (**e**). **h** Temporal ACFs of chromatin condensation dynamics measured at 10- and 20-min intervals. Dots denote the averaged data from 28 cells, shaded regions indicate standard deviations, and the solid curves represent exponential fittings.

Representative cell images of the three phases are presented in Fig. 5c. Figure 5d displays the distributions of chromatin condensation levels measured during these three phases. We observe similar distributions of chromatin condensation during the interphase, implying that the heterogeneity in chromatin condensation levels among cells is not attributable to their cell phases.

It is generally believed that the organization of chromatin changes considerably during interphase, reflecting the unique nuclear activities characteristic of each phase and their associated chromatin remodeling requirements. We point out that the variations in chromatin architecture and dynamics across different cell phases are highly dependent on the spatial and temporal scales of the measurements. For example, in the timescale of approximately 10 s, single telomeres exhibit different diffusion characteristics in the interphases where chromatin diffusion is more restricted in the S and G2 phases than in the G1 phase[57]. On the other hand, in a shorter timescale of sub-second, telomere diffusion exhibits no significant differences throughout the interphase. Another earlier study by single nucleosome tracking also reported that local chromatin motion in ~50 ms timescale is similar throughout the interphase[9]. iSCORS characterizes chromatin architectures in the short spatiotemporal scales and our data agree with these previous observations that chromatin dynamics are nearly constant throughout the interphase.

It is important to note that our findings do not suggest uniform chromatin condensation levels during interphase; instead, they indicate that chromatin condensation exhibits heterogeneity in interphase cells, which is independent of cell phases. Moreover, our study only shows that the global chromatin configuration at the small spatiotemporal scales does not change significantly during the interphase. We did not explore the local and transient chromatin reorganization which may have a closer association with specific nuclear events in the interphase, e.g., DNA replication in the S phase. Such exploration is complicated by cell movement, posing challenges to monitoring chromatin dynamics at specific chromosomes or chromosomal segments in the current setup.

## Long-term iSCORS imaging reveals temporal fluctuations in the global chromatin condensation levels within single live nuclei

Following the observation of substantial cellular heterogeneity in chromatin condensation levels among interphase cells, we set out to investigate the temporal dynamics of chromatin condensation within individual cells and its potential role in the observed cellular heterogeneity. To address this, we conduct long-term iSCORS imaging to continuously monitor chromatin condensation dynamics in individual living U2OS cells. Specifically, we measure cells at 20-min intervals, using an image acquisition rate of 1000 fps, over a period spanning up to 15 h.

Figure 6a illustrates the time-lapse chromatin condensation maps of a representative cell nucleus over a 15-h period, exhibiting considerable temporal variations during the observation period. The corresponding temporal trajectory in the $V_{DLS}$-$1/D^*$ plot displays a high correlation between $V_{DLS}$ and $1/D^*$ (as depicted in Fig. 6b), strongly suggesting that these temporal variations arise from dynamic changes in chromatin condensation levels. Figure 6c shows the temporal variation in global chromatin condensation levels of this specific cell, in which the chromatin gradually decondensed, particularly from the 9th to 15th hours of observation. It is

worth noting that other cells display uncorrelated chromatin condensation dynamics (Supplementary Fig. 15). These results indicate that the chromatin condensation levels in live cell nuclei undergo substantial temporal fluctuations, contributing to the observed cellular heterogeneity in the instantaneous snapshots.

We analyze long-term iSCORS image data for 28 cells whose time-evolving trajectories are displayed in a $V_{DLS}$-$1/D^*$ plot (Fig. 6d). Once again, $V_{DLS}$ and $1/D^*$ exhibit a strong cross-correlation (Fig. 6e), supporting that the fluctuations in $V_{DLS}$ and $1/D^*$ are primarily driven by changes in chromatin condensation levels over time. Notably, individual cells undergo heterogeneous and uncorrelated chromatin condensation dynamics with distinct fluctuation amplitudes (Fig. 6f, g). To gain a more comprehensive understanding of the chromatin condensation dynamics, we calculate the autocorrelation function (ACF) of individual cell chromatin condensation levels, followed by fitting with an exponential decay (Fig. 6h), yielding a correlation time of $51 \pm 2$ min. This characteristic time remains consistent when iSCORS imaging is conducted at a shorter time interval (every 10 min), affirming the accuracy of our measurements of chromatin condensation dynamics. Taken together, iSCORS imaging unveils spontaneous fluctuations in chromatin condensation levels of living cells, with a characteristic time scale of approximately 50 min.

## Discussion

We developed iSCORS microscopy as a label-free imaging technique to investigate chromatin organization and dynamics of live cell nuclei by measuring the DLS signal. From the DLS signal, we extracted two key features: the correlation time and fluctuation magnitude, which allowed us to deduce the nanoscopic chromatin configuration. Despite the nucleus being composed of many non-chromatin nuclear proteins and RNA, iSCORS signals predominantly originate from chromatin, evidenced by the strong correlation between iSCORS maps and fluorescence chromatin images. We attribute this to the enhanced scattering intensity of chromatin due to its dense, high-ordered structures, a feature that RNA-protein complexes lack. Occasional discrepancies between the iSCORS and fluorescence chromatin images are noted, which suggest unrelated scattering signals from non-chromatin structures, such as nuclear bodies and nuclear scaffolds. Given that the linear scattering signal is ubiquitous, achieving iSCORS signal specificity exclusively for chromatin remains a complex task. Instead, this study refrains from interpreting the subnuclear structures depicted in iSCORS maps. Our conclusions are drawn from the analysis of global chromatin condensation states, determined by calculating the spatial median intensity across the images.

Through iSCORS, we observed an enhanced chromatin condensation level upon transcription inhibition. Additionally, we measured significant temporal fluctuations and cellular heterogeneity in global chromatin condensation states. This study demonstrates a noninvasive imaging technique for studying chromatin condensation dynamics in single live cells over an extended period without the need for introducing any labels. This label-free technique provides an immediate opportunity for studying chromatin reprogramming in primary cells during processes such as stem cell differentiation and cell senescence, where fluorophore labeling is challenging. The light dose of iSCORS imaging is minimal (~0.1 kW/cm²), introducing negligible phototoxicity. Therefore, in principle, iSCORS imaging allows for indefinite observation time, although in this study, we demonstrated continuous observation over 15 h, limited by the cell culture system on our microscope. In scenarios where fluorophore labeling at gene loci and transcription machinery is feasible, simultaneous fluorescence and iSCORS imaging may enable the capture of dynamic chromatin remodeling at a specific gene during transcription in living cells.

Previous studies have conducted DLS measurements under a microscope[28,58–60], and two-dimensional mapping of the diffusion coefficient within living cells was accomplished by scanning the detection volume across the sample[29]. However, conventional DLS microscopes have not been optimized for the sensitivity and spatiotemporal resolution to detect nanoscopic molecular fluctuations in live cells. Closely related to DLS

microscopy, Fourier transform light scattering measures the spatiotemporal dynamics of a sample in the Fourier domains and is useful for studying soft matters and cellular dynamics[28,61–63]. Yet, the Fourier domain analysis produces ensemble-averaged information over the observation area without spatial resolution, making it unsuitable for studying local events. In contrast, iSCORS microscopy achieves full-field DLS imaging by employing fast-scanning laser illumination in conjunction with a high-speed camera, which significantly enhances spatiotemporal resolutions and data throughput.

The label-free iSCORS imaging enables the investigation of chromatin structures below the optical diffraction limit, thanks to the sensitivity of the DLS signal to nanoscopic organization and dynamics. However, it is important to note that the spatial resolution of iSCORS imaging, including the spatial resolutions of C, $V_{DLS}$, $D^*$, and chromatin condensation maps, is still subject to the limitations imposed by light diffraction. In iSCORS microscopy, a laser is employed to achieve a photon flux that minimizes photon shot noise when detecting chromatin DLS signals. While we use coherent laser light for illumination, we deliberately reduce the spatial coherence by tightly focusing and rapidly scanning the laser beam across the field of view. This approach effectively reduces speckle noise, thereby enhancing the spatial resolution. Future advancements may lead to even higher spatial resolution, especially in the axial direction, through the use of confocal-based interference scattering microscopy[64,65]. This enhanced resolution will improve the signal-to-noise ratios for small nuclear structures, making it essential for visualizing complex and nuclear organizations.

The current sensitivity of iSCORS microscopy is expected to be sufficient for detecting chromatin nanoclusters of ~50 nm, estimated based on a measurement noise floor of C ~0.01 and assuming a refractive index of 1.49 for the nucleosome, which closely matches the refractive index of histone core proteins. Further improvements in detection sensitivity can be achieved through back pupil function engineering, but it should be noted that higher sensitivity may require a strong light dose, which could potentially damage cells or affect nuclear activities.

In conclusion, this study reports a label-free, high-resolution 3D optical imaging technology for measuring nanoscopic chromatin arrangements in living cells. Unlike most conventional microscope techniques designed to capture the structural information of specimens, which are constrained by limitations in spatial resolution due to light diffraction, iSCORS explores nanoscopic chromatin configurations based on temporal information encoded in the DLS signal. The wealth of dynamic information recorded in the DLS signal may facilitate cross-modality image translation between label-free iSCAT imaging and other molecularly specific and functional imaging techniques[30,66].

## Methods
### iSCORS microscopy

The schematic diagram of the iSCORS microscope is displayed in Fig. 1a. A 660 nm laser beam (opus 660, Laser Quantum) was focused on the sample through a water-dipping microscope objective (OBJ1, UMPLFLN 40XW, NA0.8, Olympus). The laser was spatially scanned by two-axis acousto-optic deflectors (AODs, DTSXY-400, AA Opto) at ~100 kHz through a telescope (L1 and L2, AC254-200-A-ML and AC508-750-A-ML, Thorlabs, respectively) to create uniform illumination cross the field of view (FOV)[27]. To ensure stable illumination intensity in all images, the rapid beam scanning was synchronized with the frame acquisition through external triggers. The sample was placed at the front focal plane of OBJ1 on the stage (P-545.3C8H, Piezo System and U-780.DOS, XY stage system, Physik Instrumente). The transmitted interference signal was collected by an oil immersion microscope objective (OBJ2, UAPON 100XO, NA1.49, Olympus) and was projected on a high-speed CMOS camera (Phantom VEO 1310, Vision Research) through a camera lens L5 (AC508-750-A-ML, Thorlabs). The iSCORS images were recorded at either 5000 fps or 1000 fps, both of which provided consistent chromatin condensation results. The lower image acquisition rate at 1000 fps was employed for long-term time-lapse acquisition because it reduced the size of the dataset. The image resolution was $640 \times 640$, corresponding to a FOV of ~$27 \times 27$ µm² with a

pixel size of $42 \times 42$ nm$^2$. The illumination intensity at the sample was on the order of 0.1 kW/cm$^2$, which did not affect cell division and produced no harm to the cells. The sensitivity of the iSCORS microscope was enhanced by back-pupil function engineering, where a dot-shaped attenuator is placed in the conjugate plane of the OBJ2 back focal plane through a 4f system (L3 and L4 in Fig. 1a, AC508-200-A-ML, Thorlabs)[27]. The attenuator was custom-made, consisting of a 3 mm-diameter circular silver film deposited on a 1 mm-thick, 1-inch glass substrate by electron beam physical vapor, allowing ~1% of the light to pass through. A stage-top incubator (CU-501, Live Cell Instrument) was installed on the stage, which maintained the sample temperature and CO$_2$ concentration for live cell imaging.

For epifluorescence imaging, the sample was illuminated with a LED light (pE-300ultra) via an excitation filter F2 (FF01-405/10-25, FF01-482/25-25, or FF01-554/23-25, Semrock), lens L7 (AC254-250-A-ML, Thorlabs), and dichroic mirror DM (Di01-R405/488/561/635-25). The excitation filters were installed in pE-300 ultra for different channels. The fluorescence image was recorded by EMCCD (iXon Ultra 897, Andor) through lens L6 (AC508-300-A-ML, Thorlabs), and emission filter F1 (FF01-452/45-25, FF01-520/35-25, or FF01-600/37-25, Semrock). The emission filters were set in the filter wheel (FW103H/M, Thorlabs) and can be switched for different fluorescence channels.

For automated long-term measurements, a home-written Python code was written to coordinate the video acquisitions and stage positioning.

### Cell culture and cell imaging
U2OS cells were obtained from ATCC and cultured in Dulbecco's Modified Eagle Medium (DMEM, HyClone) supplemented with 10% fetal bovine serum (FBS, HyClone), L-glutamine, and 1% penicillin/streptomycin. To visualize chromatin in live cells, U2OS cells were genetically modified by a knock-in technique using a plasmid of H2B-mCherry (pH2B_mCherry_IRES_neo3, Addgene, #21044) and generated a stable cell line (Omisc Bio, Taiwan). U2OS H2B-mCherry cell line was maintained in DMEM supplemented with 10% FBS, L-glutamine, and 400 ng/mL G418 (Gibco, #10131027). HeLa Fucci (CA2) cells were cultured in Minimum Essential Medium (MEM, HyClone, #SH30265.FS) with 10% FBS and 1% penicillin/streptomycin. All of the cells were incubated in a 5% CO$_2$ incubator (Astec, Inc.) at 37 °C. Cells were seeded on 35 mm coverglass-bottom dishes or the dishes with grid markers (GWST-3512, WillCo-dish or μ-Dish 35 mm, high Grid-500 Glass Bottom, ibidi) and grew at 70% confluency before imaging. For long-term live cell observation, the cells were maintained in a stage-top micro-incubator at 37 °C and 5% CO$_2$. Cells were replaced with fresh culture medium before imaging.

### Drug treatments
We followed the conditions of drug treatments reported in previous studies that effectively modify the chromatin compaction or inhibit the gene transcription without introducing noticeable apoptosis. When applicable, we intend to shorten the drug treatment duration to further reduce the long-term side effects on the cells. To induce global chromatin decompaction, cells were treated with 5 mM sodium butyrate (NaB)[67] or 500 nM Trichostatin A (TSA)[68] for 2 h. To induce global chromatin condensation, 1 μM Actinomycin D (ActD) was incubated with the cells for 2 h[17]. For the ATP depletion, 10 mM sodium azide and 50 mM 2-deoxyglucose were added to the cells and incubated for 30 min[49]. For transcription inhibition, 100 μM α-AM[69] or 200 μM DRB[9] was incubated with cells for 2 h.

### Immunostaining and RNA labeling
Cells were washed with PBS and penetrated with 0.1% Triton X-100 in PBS for 10 min. Then, cells were fixed with 4% paraformaldehyde at RT for 10 min. To label the nuclear speckle, cells were incubated with blocking buffer (1% BSA/PBS) at room temperature for 30 min and labeled with Anti-SON antibody (abcam, #ab121759, dilution 1:200) for 2 h at room temperature. After primary antibody incubation, cells were further incubated with secondary antibody (Thermo Fisher Scientific, Alexa Fluor 647 goat anti-rabbit IgG, dilution 1:1000) for an hour at room temperature. To

visualize RNA, fixed cells were incubated with 5 nM SYTO RNASelect Green Fluorescent Cell Stain (Invitrogen, #S32703) for 15 min at room temperature and washed with PBS before imaging.

### Calculation of the D* map and V$_{DLS}$ map
The raw image was performed by $4 \times 4$ binning to reduce the photon noise. ACF and temporal variance were calculated from a 1-s video. Multiple videos were recorded consecutively, and their V$_{DLS}$ and D* maps were averaged to improve measurement accuracy. The data presented in this study represent averages of 18 videos, each lasting one second. The analysis was implemented using custom scripts written in MATLAB (MATLAB R2018b, MathWorks).

### Quantification of chromatin condensation level by iSCORS
The level of chromatin condensation is defined by the distance between the iSCORS data point and the noise baseline data point, projected onto a line in the log-log plot of V$_{DLS}$ and 1/D*. For this assessment, the noise baselines for V$_{DLS}$ and 1/D* are set at $10^{-5}$ and $10^{-3}$, respectively. The calculated condensation level does not depend on the choice of the $y$-intercept value of the line being projected.

### Nucleus segmentation
The region of the cell nucleus was identified in the C map using the machine learning model from our previous study[25]. The results were visually inspected, and minor manual corrections were made when necessary.

### DNA staining and calculation of CV
The method was previously described[10]. To access the chromatin condensation in live cells, the nucleus was stained with Hoechst 33420 in the culture medium for 10 min at 37 °C, 5% CO$_2$ incubator before imaging. The mid-section of a nucleus was imaged using confocal microscopy (Eclipse Ti2, Nikon with CSU-X1, Yokogawa). CV was determined by dividing the standard deviation of the nuclear fluorescence intensity by the mean nuclear fluorescence intensity.

### Statistics and reproducibility
Sample sizes and $P$ values are denoted in the figure legends. Statistical analyses were performed and displayed as box plots with Origin Pro 2021. Statistical significance was determined by comparing the experimental group with the control group, and an unpaired two-tailed Student's $t$ test was used. In the box plots, the lower and upper boundaries of the box indicate the 25th and 75th percentiles, respectively. Within the box, a solid line marks the median, and a large white circle marks the mean. Whiskers above and below the box indicate the mean ± SD, and small dots represent the data points of individual cells. [***$P < 0.001$ (Student's $t$ test)].

## Data availability
iSCORS image data of a representative cell nucleus are available in the Zenodo database (https://zenodo.org/records/11182995)[70]. Numerical source data for the main graphs are provided as Supplementary Data. Other supplementary data in this study are available from the corresponding author upon reasonable request.

## Code availability
Relevant codes for iSCORS analysis have been deposited on Zenodo (https://zenodo.org/records/11182995)[70].

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

## Acknowledgements
This work is supported by Academia Sinica, Taiwan (AS-iMATE-111-35 & AS-GCS-113-M03) and National Science and Technology Council (NSTC), Taiwan (NSTC 111-2112-M-001-051-MY5). We acknowledge support by the Biophysics Core Facility at the Institute of Atomic and Molecular Sciences, Academia Sinica. We thank the National Center for High-performance Computing (NCHC) of National Applied Research Laboratories (NARLabs) in Taiwan for providing computational and storage resources.

## Author contributions
C.-L.H. and H.-P.C.C. conceived the idea for the project and designed the experiments. Y.-T.H. established the optical microscope and image data analysis for iSCORS imaging under the supervision of C.-L.H. I.-H.L. and B.-K.W. prepared the cell samples. I.-H.L. established the CV measurement under the supervision of C.-L.H. Y.-T.H., I.-H.L., and B.-K.W. performed the experiments and analyzed the data. The manuscript was prepared by Y.-T.H., I.-H.L. and C.-L.H. All authors contributed to this work, discussed results and conclusions, and commented on the manuscript.

## Competing interests
The authors declare no competing interests.
