## [Peer review file · Communications Biology]

Reviewers' comments:

Reviewer #1 (Remarks to the Author):

In their manuscript titled 'Probing Chromatin Condensation Dynamics in Live Cells Using Interferometric Scattering Correlation Spectroscopy,' Hsiao et al. introduce a novel scattering-based optical microscopy technique. This method facilitates the probing of chromatin organization and dynamics in live cells by analyzing fluctuations and (de)correlation of linear scattering signals. Utilizing this label-free technique, the authors successfully reveal nano-structures of chromatin, detect native chromatin motion, and map the spatial distribution of condensed chromatin at the nanoscale with sub-millisecond resolution. Their findings include spontaneous fluctuations and heterogeneous compaction levels in chromatin. Additionally, they observe a correlation between chromatin condensation and transcription inhibition, but not with specific interphase phases. Despite the significant insights and claims presented, the underlying principles and findings of this study require further validation. Nevertheless, this work represents an intriguing advancement in iSCAT (interferometric scattering) microscopy, drawing inspiration from dynamic light scattering and correlation spectroscopy. This novel approach shows promise for investigating biological systems dynamically. While the results are both impressive and interesting, there are several aspects that require further clarification to reinforce the manuscript's suitability for publication in this journal.

(1) The authors measured the D^* of chromatin based on the ACF_DLS, the functional form of which is given in Eq (5). While the ACF might physically take an exponential form, this manuscript does not provide a clear derivation, justification, or convincing argument to confirm this assumption. Given that the entire analysis of this work is based on this formula, its validity needs to be thoroughly established.

(2) Building on point (1), where the authors measure the D^* of chromatin based on ACF_DLS as described in Eq (5), it is crucial to address a potential inconsistency. Eq (5) appears to represent a formula for homodyne Dynamic Light Scattering (DLS), yet iSCAT, as employed in this study, is not a homodyne technique since the scattered field interferes with the reference field from the same beam path. Could the authors provide an explanation and justification for the application of this equation in the context of iSCAT?

(3) Continuing from point (1), I have a question regarding the applicability of Eq (5) to anomalous subdiffusion. The authors have stated that chromatin undergoes anomalous subdiffusion, yet Eq (5) is traditionally associated with normal diffusion. Could the authors clarify whether this equation is appropriate for describing the anomalous subdiffusion observed in chromatin?

(4) The definition of C_{DLS} presented in the manuscript is somewhat confusing. Specifically, there is a concern that C_{DLS} might diverge in instances where τ becomes very small after averaging. This issue seems to be connected to the selection of the observation time used to calculate τ . Could the authors provide a clearer explanation regarding how observation time is chosen and its impact on the calculation of C_{DLS} ?

(5) Figure 1(d) illustrates that the chromatin boundary exhibits a low D^* , suggesting reduced fluctuation levels at this boundary. Conversely, Figure 1(e) indicates significant temporal intensity variance at the boundary. The authors propose that an anti-correlation between D^*

and V_{DLS} (low D^* correlating with high V_{DLS}) is logical. However, an alternative theory could be that regions with low D^* are less mobile and more stationary, thus exhibiting fewer fluctuations and resulting in a smaller V_{DLS} . Could the authors please provide their insights on this perspective?

(6) This manuscript seems to be closely related to, and possibly an extension of, the authors' previous work published in ACS Nano in 2022. Could the authors clarify whether there are any differences in the experimental setup compared to the earlier study? Additionally, are there any improvements in axial resolution? If so, what are the reasons behind these enhancements? Given the close relationship between these studies, it would be beneficial for the authors to explicitly discuss the similarities and differences between the previous work and this one in the introduction.

(7) In Figure 2(a), the authors compare chromatin condensation maps with corresponding fluorescent images, highlighting the superior spatial resolution of the former. However, a more appropriate comparison might be with confocal fluorescence images or super-resolution images, as these modalities offer enhanced spatial resolution in fluorescence imaging. I suggest the authors address this point to provide a fairer comparison and justification of their method's advantages.

(8) In Figure 3, where the authors present optically sectioned images obtained through iSCORS microscopy, providing 3D reconstructed images of chromatin organization would enhance the readers' understanding. Such images could offer additional insights into the spatial arrangement and structural complexities of chromatin.

(9) The authors note in Figure 5 that the level of chromatin condensation remains nearly constant throughout the cell cycle. This finding is both unexpected and surprising. If this is a novel discovery made in the current work, it warrants greater emphasis for clarity in the manuscript. Alternatively, if this observation has been previously reported, appropriate citations to the original work are necessary.

(10) Figure S10, which illustrates chromatin organization and dynamics during a cell's mitosis, is particularly interesting. It clearly shows a dramatic change in chromosome organization. However, despite its significance, this result is neither thoroughly described nor analyzed in the manuscript. Have various phases of mitosis, such as metaphase and anaphase, been observed? Additionally, the images appear quite noisy. Could this be the reason for their exclusion from the main text figures? An explanation would be beneficial.

Overall, this study presents both interesting and valuable contributions to the field. However, to ensure its suitability for publication, it is essential that the aforementioned issues are comprehensively addressed and resolved.

Reviewer #2 (Remarks to the Author):

This manuscript is an extension of the author's previous work published in ACS Nano (PMID: 34967599), in which the label-free imaging (interferometric scattering or iSCAT) was justified. The authors further optimized iSCAT by analyzing dynamic light scattering (DLS) signals by using

a correlation spectroscopic analysis that measures the diffusion coefficient and density of chromatin. The new method is named interferometric scattering correlation spectroscopy (iSCORS). This work probed the dynamics and heterogeneity of chromatin condensation. While I found this work interesting and consider iSCORS to have the potential to make great discoveries in live cells, I am not convinced that the signals detected by iSCORS merely represent chromatin. DNA and RNA are highly similar in chemical compositions (ATCG vs AUCG). Besides highly concentrated RNA in nucleoli, RNA can be detected everywhere in the nucleus in the form of protein-RNA complexes. RNA and proteins can be captured at a fast acquisition rate. This work used 1000 or 5000 fps, which is fast enough to capture RNA and protein movement in the nucleus. In addition, RNA can form granules and participate in chromosome compaction. How RNA and nuclear proteins contribute to the signals in this work has not been discussed or calibrated. The diffusion constants of chromatin measured by iSCORS are much faster than those reported by other groups which can probably be explained by contributions from freely diffusing particles. The authors should address this issue to be considered by the journal.

The second major concern is the conclusion of chromatin condensation during transcription inhibition by DRB. It is known that inhibition of pol II activity induced redistribution of nucleolar proteins around the nucleoli. Such phenomenon is driven by RNA and nucleolar proteins, not chromatin. The figures shown in this manuscript look like the reorganization of nucleoli, which could serve as evidence that iSCORS does not distinguish chromatin from RNA-protein complexes. The authors should provide data from parallel approaches with direct evidence to strengthen the conclusions.

The heterogeneity of chromatin organization in the nucleus has been shown by other groups. The authors should include more references to clarify the existing knowledge and the significance of their discovery. Other than the above concerns, suggestions with more details are listed below:

Comments:

1. The major concern for this manuscript is that the nucleus is composed of many molecules – DNA, RNA, protein, and metabolites. Although the authors showed a correlation between the VDLS map and the H2B-mCherry image, inconsistent regions can still be found, such as certain bright spots in the H2B-mCherry aligned with dark regions in the VDLS map and vice versa. An interesting question is whether the inconsistent regions reflect nucleosome-free DNA (open chromatin) or RNA condensates/clusters. I recommend the authors provide correlation data of their VDLS map with a live-cell DNA stain, such as Vybrant DyeCycle Ruby Stain, which labels the entire genome in a live setting.
2. The reported apparent diffusion coefficient ($0.92 \pm 0.07 \mu\text{m}^2/\text{s}$, line 164) is approximately 10 times larger than the reported mRNA diffusion constants in the nucleus (PMID: 12546792). Given that RNA is less constrained than chromatin, an explanation for this in the discussion would help the readers understand the meaning of these parameters.
3. How the effective concentration of each treatment was determined is not clear. Such information is important because some reagents showed additional functions (e.g., triggering apoptosis) when used in a high concentration. Please include this information in the manuscript.

4. Treatment of DRB does not have direct effects on chromatin states. DRB inhibits transcription by inhibiting RNA elongation of RNA pol II, which stops the synthesis of new RNA molecules. However, DRB neither degrades the existing RNA nor alters chromatin states (considering chromatin compaction is primarily regulated by histone modifications and SMC complexes.) The reference cited (#41) confirmed that treatment of DRB alone has no obvious change to chromatin compaction. The chromatin condensation observed by the authors (line 277) during DRB treatment is mostly likely changes in RNA or protein, not DNA. The authors should simultaneously use a live-cell DNA stain (e.g., Vybrant DyeCycle Ruby Stain) and a live-cell RNA stain (e.g., SYTO) to strengthen their conclusions and explain the rationale of their results.

5. In line 293, the authors claimed that “The observed chromatin condensation dynamics induced by DRB treatment, depicted in Fig 4g, providing clear evidence of chromatin condensation...” This statement needs additional evidence from DNA specific experiments.

6. The authors mentioned the results that chromatin in mammalian cells undergoes anomalous subdiffusion. In addition to single nucleosome tracking (ref [14, 15] proving chromatin dynamics averaged across the whole nucleus, similar results were also reported by other approaches, such as LaO and CRISPR-based imaging that can probe the dynamics of specific genomic loci.

7. The authors propose two quantities (VDLS, D^*) to characterize chromatin condensate dynamics. The log-log plot of these two quantities shows a line with slope 3 followed by the dependence of VDLS $\sim (1/D^*)^3$. What is the meaning of the y-intersect on the log-log plot? Is it a cell-type-dependent constant or a chromatin-location-dependent constant? The authors should discuss it.

8. Related to the comment above, in Fig 2b, the authors performed the perturbation of global chromatin condensation by chemicals. These data seem not best fitted into the same line with a slope 3 but into parallel lines with the same slope 3 but different y-intercepts. The authors are encouraged to discuss the meaning of these discrepancies.

9. In Figure 2c, the authors should explain how the chromatin condensation levels were calculated. The authors indicated that chromatin condensation levels were generated by the projection of data points on the line of slope 3. Are the data points projected on the same line? After projection, how to read off the values (for chromatin condensation level)? The authors should explain why this value can be used to characterize the level of chromatin condensation.

10. The authors observed similar distributions of chromatin condensation during the interphase. The authors should discuss the findings using other approaches, such as global chromatin domain analysis and individual chromatin compactations.

Minor comments:

11. Small grammar issues, such as a missing period in line 87, should be fixed.

12. Reference 8 has duplicated “doi:.” One of them should be removed.

13. The nucleus boundary in Figure 1d and 1e does not fit the nuclear boundary in 1f. Please provide the information regarding how the nuclear boundary was determined.

14. The source of fluorescence in the fluorescence images in Figure 2a is unclear. Please provide the information in the figure legend.
15. Two graphs are labeled as Figure 5c in the legend.
16. Please include the number of cells in Figure 5d.
17. The information regarding the source of the U2OS H2B-mCherry cells is missing.

Reviewer #3 (Remarks to the Author):

In this manuscript, Hsiao et al. have developed a label-free bright-field microscopic method, termed iSCORS, to investigate chromatin condensation in living cells. This method extends the previously reported interferometric scattering microscopy technique. The spatial resolution is a similar range to the spinning-disk confocal microscopy. Chromosome condensation levels were assessed by obtaining the temporal variance of dynamic light scattering (VDLS), the apparent diffusion coefficient (D^*), and the interference contrast (C). Control experiments using drugs known to trigger chromosome condensation (actinomycin D, 2-deoxyglucose, and sodium azide) and decondensation (trichostatin A and sodium butyrate) demonstrated that VDLS, $1/D^*$, and C could measure chromosome condensation levels. Upon treatments with transcription inhibitors, chromatin condensation levels were increased. The condensation induced by DRB was reversed upon its removal. During the cell cycle, chromosome condensation levels were relatively constant during the G1 and S phases, while being slightly decreased during the G2.

The technique is unique and potentially powerful for understanding chromatin structure at the nanoscale level in living cells without fluorescence labels. However, the results provide little biological advance. The label-free technique should offer advantages such as high-speed and less-toxic imaging at high resolution, but none of these were explored. I believe it is essential to demonstrate findings that are exclusively obtainable using this new technique for publication in a high journal.

Specific points

1. Although there are some intriguing data that may contradict previous findings, an orthogonal approach is essential for validating the conclusion. It has been shown that DRB treatments do not induce chromatin condensation and rather induce decondensation (e.g., doi.org/10.1006/excr.1996.0124). In addition, chromatin condensation induced by actinomycin D has been detected in many studies using GFP-tagged histones and Hoechst staining, unlike stated in Introduction and shown by the authors. The analytical method involving edge detection used in the manuscript may not be suitable to assess the chromatin compaction. Consider using another method, such as the standard deviation-based measurements.
2. Although the iSCORS technique has a high spatial resolution, the analyses are all the average of a single nucleus and the outcome is not very novel. Since local chromatin condensation levels change during DNA replication, the technique could be used to demonstrate such changes during the S phase. It would be interesting if the temporal order of replication from euchromatic to heterochromatic regions is detected by iSCORS.

We appreciate the valuable feedback and recommendations from all reviewers. Below are detailed responses to each comment (in blue). The corresponding changes made to the main text are included in this response document, marked in green. Additionally, a version of the manuscript that highlights these changes has been uploaded to the submission system.

Reviewer #1 (Remarks to the Author):

In their manuscript titled 'Probing Chromatin Condensation Dynamics in Live Cells Using Interferometric Scattering Correlation Spectroscopy,' Hsiao et al. introduce a novel scattering-based optical microscopy technique. This method facilitates the probing of chromatin organization and dynamics in live cells by analyzing fluctuations and (de)correlation of linear scattering signals. Utilizing this label-free technique, the authors successfully reveal nano-structures of chromatin, detect native chromatin motion, and map the spatial distribution of condensed chromatin at the nanoscale with sub-millisecond resolution. Their findings include spontaneous fluctuations and heterogeneous compaction levels in chromatin. Additionally, they observe a correlation between chromatin condensation and transcription inhibition, but not with specific interphase phases. Despite the significant insights and claims presented, the underlying principles and findings of this study require further validation. Nevertheless, this work represents an intriguing advancement in iSCAT (interferometric scattering) microscopy, drawing inspiration from dynamic light scattering and correlation spectroscopy. This novel approach shows promise for investigating biological systems dynamically. While the results are both impressive and interesting, there are several aspects that require further clarification to reinforce the manuscript's suitability for publication in this journal.

(1) The authors measured the D^* of chromatin based on the ACF_DLS, the functional form of which is given in Eq (5). While the ACF might physically take an exponential form, this manuscript does not provide a clear derivation, justification, or convincing argument to confirm this assumption. Given that the entire analysis of this work is based on this formula, its validity needs to be thoroughly established.

We agree with the reviewer that it is important to justify the utilization of Eq. (5) in fitting our experimental data. Comments #2 and #3 are closely related, and we place our responses to the comments on interferometric DLS measurement and chromatin subdiffusion separately in the later sections. Modifications in the main text have been made in response to comments #1 to #3.

Eq. (5) has been derived based on the prior studies of DLS modeling for free diffusion of nanoparticles. Using nanoparticle samples, we have verified that iSCORS data matches with the DLS model, as illustrated in Supplementary Fig. 4. For chromatin dynamics in living cells, we reason that the movement of chromatin at the nanoscale is dominated by thermal fluctuation, and therefore follows the physical rules of diffusion. This is supported by the exponential decay of the ACF measured from live cell nuclei in the short timescale of milliseconds (data shown below). We note that chromatin undergoes subdiffusion, instead of free diffusion as considered in the DLS model of Eq. (5). In our response to comment #3, we discuss the applicability of using apparent

diffusion coefficient D^* at a specific timescale to characterize the chromatin subdiffusion based on the DLS model.

Representative DLS signal of a live cell nucleus (top) and its autocorrelation function with an exponential fitting (bottom).

(2) Building on point (1), where the authors measure the D^* of chromatin based on ACF_DLS as described in Eq (5), it is crucial to address a potential inconsistency. Eq (5) appears to represent a formula for homodyne Dynamic Light Scattering (DLS), yet iSCAT, as employed in this study, is not a homodyne technique since the scattered field interferes with the reference field from the same beam path. Could the authors provide an explanation and justification for the application of this equation in the context of iSCAT?

DLS models can be categorized into two detection modes: homodyne and heterodyne, distinguished by the absence or presence of a coherent reference beam, respectively. In our study, the iSCORS employs the heterodyne mode, characterized by the dominance of the time-varying interference signal created by the interference of a strong reference beam with the scattering signals from the cell sample.

It is worth noting that, in both the homodyne and heterodyne modes, the autocorrelation functions of the DLS signals have an exponential form but their decay rates are different by a factor of two (please see the detailed derivations in the Supplementary Note added in this revision). With proper calibrations with nanoparticle colloids, Eq. (5) is valid for the DLS model of the heterodyne mode.

We have supplemented our revised manuscript with detailed derivations of both homodyne and heterodyne DLS detections in the Supplementary Note to further substantiate our methodology and findings.

In the main texts, we have added:

Page 5: “Specifically, our iSCORS imaging resembles a DLS measurement with a coherent reference beam.³³ Based on the DLS modeling, we estimate D^* by calculating the temporal autocorrelation function (ACF) of the contrast fluctuation of every pixel.”

Page 5: “By calibrating the system with nanoparticle colloids of known diffusion coefficients, we are able to measure the D^* of chromatin based on the ACF (see derivation in Supplementary Note and data in Supplementary Fig. 4).

”

(3) Continuing from point (1), I have a question regarding the applicability of Eq (5) to anomalous subdiffusion. The authors have stated that chromatin undergoes anomalous subdiffusion, yet Eq (5) is traditionally associated with normal diffusion. Could the authors clarify whether this equation is appropriate for describing the anomalous subdiffusion observed in chromatin?

Indeed, several previous studies have reported that chromatin movement is subdiffusive, a characteristic that exhibits a decreasing apparent diffusion coefficient as the measurement timescale increases (DOI: 10.1091/mbc.E14-06-1127; DOI:10.1126/sciadv.abn5626; DOI:10.1073/pnas.1907342116). Due to the limited dynamic range of timescale a measurement could typically cover (especially for fluorescence-based measurement where the observation time is restricted by the photobleaching effect), it is informative to characterize the subdiffusion by measuring the apparent diffusion coefficient (D^*) at specific timescales.

We show that by down-sampling the iSCORS data in time, we can measure a decreasing D^* as the timescale increases, corresponding to the chromatin subdiffusion in the millisecond timescales (data shown below).

The apparent diffusion coefficient D^ diminishes with increasing measurement timescales, indicating the subdiffusive behavior of chromatin within the nucleus of live cells.*

In the study of chromatin condensation, we choose to use the D^* at the shortest timescale of our measurement (1 ms) because it is most sensitive to chromatin condensation as it mainly measures the local fluctuation of chromatin driven by thermal energy. As a demonstration, we show that the ability of iSCORS to determine chromatin condensation diminishes significantly when operating at a longer timescale of 10 ms (data shown below, in comparison to the data of 1 ms in Fig. 2b). This may be because the DLS signals at longer timescales are affected more by the macroscopic properties of chromatin instead of the nanoscopic chromatin compaction states. Therefore, employing D^* at the short timescale, we effectively assess chromatin condensation levels, underpinning our methodological choice with both theoretical and empirical evidence for the robust and appropriate use of Eq. (5) in our analysis.

The iSCORS data was analyzed at a longer timescale of 10 ms for cells treated with chemical drugs that alter the global chromatin condensation levels. Compared to the data of 1 ms, the distinctions between the different treatments are less pronounced in the V_{DLS} - $1/D^*$ graph of the 10-ms data, indicating reduced sensitivity to changes in chromatin condensation states.

Page 5: “Early studies show that chromatin in mammalian cells undergoes anomalous subdiffusion, measured by tracking single nucleosome and specific genomic loci.^{14,15,31,32} In this study, we choose to measure an apparent diffusion coefficient, denoted as D^* , at the shortest timescale of iSCORS measurement because of its high sensitivity to the nanoscopic chromatin configuration. In the timescale of our iSCORS imaging at sub-milliseconds, chromatin movement is on the nanometer scale that corresponds to local diffusion driven by thermal fluctuation. Thus, we reason that, by analyzing the iSCORS signal with DLS models, the D^* of chromatin at the nanoscale can be determined.”

Page 5: “We confirm this phenomenon by increasing the measurement timescale of iSCORS to 5 ms, and indeed a significantly reduced D^* is measured, consistent with the behavior of chromatin subdiffusion (Supplementary Fig. 5).”

(4) The definition of C_{DLS} presented in the manuscript is somewhat confusing. Specifically, there is a concern that C_{DLS} might diverge in instances where τ becomes very small after averaging. This issue seems to be connected to the selection of the observation time used to calculate τ . Could the authors provide a clearer explanation regarding how observation time is chosen and its impact on the calculation of C_{DLS} ?

Selecting an optimal observation window is crucial for accurately capturing the DLS signals associated with chromatin fluctuations, balancing the need to gather sufficient statistical data without including nonspecific scattering caused by cellular movement. Guided by this principle, we determined that an observation duration of 1 second, coupled with an image acquisition rate of 1000 Hz, keeps the right balance.

We have conducted a quantitative analysis of how the length of the observation window impacts the precision of background intensity estimation (data below). It reveals that windows extending beyond approximately 100 ms, encompassing over 100 frames, are effective for reasonable

background assessments. Concurrently, to mitigate the influence of irrelevant background variations resulting from cellular activities, we avoided extending the observation period beyond 5 seconds.

The time-averaged background was calculated over various observation windows. The longest window, 3.5 seconds, was considered as the optimal background estimation. Differences between the backgrounds calculated at shorter timescales and the optimal estimation were depicted (in panel a) and quantified (in panel b). The results indicate that the background estimation becomes more consistent as the observation window lengthens, with the 1-second window offering a reasonably accurate estimation.

Page 4: “This choice of observation time of 1 second is a compromise to ensure sufficient statistical data is gathered for calculating the time average while excluding scattering signal changes that arise from cell movements and are not relevant to chromatin dynamics.”

(5) Figure 1(d) illustrates that the chromatin boundary exhibits a low D^* , suggesting reduced fluctuation levels at this boundary. Conversely, Figure 1(e) indicates significant temporal intensity variance at the boundary. The authors propose that an anti-correlation between D^* and V_{DLS} (low D^* correlating with high V_{DLS}) is logical. However, an alternative theory could be that regions with low D^* are less mobile and more stationary, thus exhibiting fewer fluctuations and resulting in a smaller V_{DLS} . Could the authors please provide their insights on this perspective?

V_{DLS} is sensitive to the mass density in motion by quantitating the amplitude of fluctuations in the DLS signal, which is distinct from the concept of mobility (D^*) that is determined by the correlation time of the DLS signal. In the ACF analysis of DLS signal, V_{DLS} represent the value of the first data point in the ACF, whereas D^* represents the decay rate of the ACF. While the assessments of V_{DLS} and D^* operate on separate theoretical foundations, their relevance emerges when examining the dynamics of cell chromatin, attributed to the inherent physical characteristics of chromatin. Our experimental findings, highlighted in Figure 2, demonstrate an inverse relationship between V_{DLS} and D^* , where chromatin, when condensed due to the action of transcription inhibitors or ATP depletion, shows increased V_{DLS} values alongside reduced D^* . This pattern underscores the elevated molecular compaction of chromatin and its consequent diminished mobility.

(6) This manuscript seems to be closely related to, and possibly an extension of, the authors' previous work published in ACS Nano in 2022. Could the authors clarify whether there are any differences in the experimental setup compared to the earlier study? Additionally, are there any improvements in axial resolution? If so, what are the reasons behind these enhancements? Given the close relationship between these studies, it would be beneficial for the authors to explicitly discuss the similarities and differences between the previous work and this one in the introduction.

The main improvement of the optical system in this study compared to our previous demonstration is the higher axial resolution. In iSCORS, the spatial resolution is determined both by the NAs of the condenser lens and the imaging objective lens. In this study, we enhanced the resolution by replacing the original low-NA condenser lens with a high-NA condenser lens (40x NA0.8 water-immersion objective). The axial resolution in our previous study was $\sim 2.75 \mu\text{m}$, whereas in this study, the axial resolution is notably improved, measuring $\sim 1.1 \mu\text{m}$ (Supplementary Fig. 12).

Page 3: "Additionally, we have upgraded our earlier system by incorporating a high numerical aperture microscope condenser for laser illumination. This enhancement has allowed us to achieve three-dimensional (3D), label-free imaging of chromatin, significantly improving spatial resolution."

(7) In Figure 2(a), the authors compare chromatin condensation maps with corresponding fluorescent images, highlighting the superior spatial resolution of the former. However, a more appropriate comparison might be with confocal fluorescence images or super-resolution images, as these modalities offer enhanced spatial resolution in fluorescence imaging. I suggest the authors address this point to provide a fairer comparison and justification of their method's advantages.

We have replaced the epifluorescence images with confocal fluorescence images by performing new experiments, which offer higher spatial resolutions and the results are more consistent with iSCORS images (see the new data below). To acquire iSCORS and confocal fluorescence images of the same cell, we culture cells on a glass substrate with grid markers so that the same cell can be identified under two microscope systems. Cells are chemically fixed immediately after iSCORS measurement, and then fluorescence confocal imaging is performed on fluorophore-stained cells. This set of new image data shows that the higher resolutions in the fluorescence confocal images help to evaluate the similarities between the fluorescence chromatin image and the iSCORS image data.

We believe that the main advantages of iSCORS lie in its label-free nature, saving the efforts for fluorophore attachment and completely avoiding potential labeling artifacts (e.g., the toxicity introduced by DNA stains). iSCORS also facilitates long-term observations without the limitations of photobleaching. Finally, iSCORS measures simultaneously the chromatin mobility and nanoscopic compaction states that are difficult to characterize by conventional fluorescence microscopes.

We have revised Figures 1g and 2a and highlighted the advantages of iSCORS in the main text.

Revised Figure 1e-1g

Revised Figure 2a

Page 8: “Compared to these fluorescence-based methods, the significance of iSCORS imaging lies in its label-free nature, thus facilitating the observation of chromatin remodeling within native live cells, devoid of the potential perturbations that fluorophore labeling may introduce.”

(8) In Figure 3, where the authors present optically sectioned images obtained through iSCORS microscopy, providing 3D reconstructed images of chromatin organization would enhance the readers' understanding. Such images could offer additional insights into the spatial arrangement and structural complexities of chromatin.

We thank the reviewer for the suggestion. We have made 3D visualizations for the chromatin condensation maps and corresponding fluorescence confocal images, including new figures in Figure 3 (copied below) and a supplementary movie.

Revised Figure 3

(9) The authors note in Figure 5 that the level of chromatin condensation remains nearly constant throughout the cell cycle. This finding is both unexpected and surprising. If this is a novel discovery made in the current work, it warrants greater emphasis for clarity in the manuscript. Alternatively, if this observation has been previously reported, appropriate citations to the original work are necessary.

It is generally believed that the organization of chromatin changes considerably during interphase, reflecting the unique nuclear activities characteristic of each phase and their associated chromatin remodeling requirements. We point out that the variations in chromatin architecture and dynamics across different cell phases are highly dependent on the spatial and temporal scales of the measurements. For example, in the timescale of approximately 10 seconds, single telomeres exhibit different diffusion characteristics in the interphases where chromatin diffusion is more restricted in the S and G2 phases than in the G1 phase (DOI: 10.1016/j.isci.2022.104197). On the other hand, in a shorter timescale of sub-second, telomere diffusion exhibits no significant differences throughout the interphase. Another earlier study by single nucleosome tracking also reported that local chromatin motion in ~50 ms timescale is similar throughout the interphase (DOI: 10.1016/j.molcel.2017.06.018) These previous observations are supported by the iSCORS method that characterizes chromatin fluctuation in the short spatiotemporal scales.

We emphasize that our study only shows that the global chromatin configuration at the small spatiotemporal scales does not change significantly during the interphase. We did not explore the local and transient chromatin reorganization which may have a closer association with specific nuclear events in the interphase, e.g., DNA replication in the S phase. Such exploration is complicated by cell movement, posing challenges to monitoring chromatin dynamics at specific chromosomes/chromosomal segments.

Page 10: “It is generally believed that the organization of chromatin changes considerably during interphase, reflecting the unique nuclear activities characteristic of each phase and their associated chromatin remodeling requirements. We point out that the variations in chromatin architecture and dynamics across different cell phases are highly dependent on the spatial and temporal scales of the measurements. For example, in the timescale of approximately 10 seconds, single telomeres exhibit different diffusion characteristics in the interphases where chromatin diffusion is more restricted in the S and G2 phases than in the G1 phase.⁵⁶ On the other hand, in

a shorter timescale of sub-second, telomere diffusion exhibits no significant differences throughout the interphase. Another earlier study by single nucleosome tracking also reported that local chromatin motion in ~50 ms timescale is similar throughout the interphase.⁹ iSCORS characterizes chromatin architectures in the short spatiotemporal scales and our data agree with these previous observations that chromatin dynamics are nearly constant throughout the interphase.”

(10) Figure S10, which illustrates chromatin organization and dynamics during a cell's mitosis, is particularly interesting. It clearly shows a dramatic change in chromosome organization. However, despite its significance, this result is neither thoroughly described nor analyzed in the manuscript. Have various phases of mitosis, such as metaphase and anaphase, been observed? Additionally, the images appear quite noisy. Could this be the reason for their exclusion from the main text figures? An explanation would be beneficial.

During mitosis, cell structure undergoes major changes, including chromatin condensation and the breakdown of the nuclear envelope. As a result, we expect the specificity of DLS signals associated with chromatin to decrease during this phase. Additionally, the analysis of DLS data on chromatin condensation relies on the assumption of uniform chromatin density, which may not be valid during mitosis. Therefore, we are careful with our interpretation of DLS measurements during mitosis, opting to note the increase in molecular density without extensive analysis.

Given these considerations, we believe that the present data and its analysis are not sufficiently mature for a detailed examination of chromatin condensation in the context of mitosis. To prevent any misinterpretation, we have chosen to remove these data from the main text.

Overall, this study presents both interesting and valuable contributions to the field. However, to ensure its suitability for publication, it is essential that the aforementioned issues are comprehensively addressed and resolved.

Reviewer #2 (Remarks to the Author):

This manuscript is an extension of the author's previous work published in ACS Nano (PMID: 34967599), in which the label-free imaging (interferometric scattering or iSCAT) was justified. The authors further optimized iSCAT by analyzing dynamic light scattering (DLS) signals by using a correlation spectroscopic analysis that measures the diffusion coefficient and density of chromatin. The new method is named interferometric scattering correlation spectroscopy (iSCORS). This work probed the dynamics and heterogeneity of chromatin condensation. While I found this work interesting and consider iSCORS to have the potential to make great discoveries in live cells, I am not convinced that the signals detected by iSCORS merely represent chromatin. DNA and RNA are highly similar in chemical compositions (ATCG vs AUCG). Besides highly concentrated RNA in nucleoli, RNA can be detected everywhere in the nucleus in the form of protein-RNA complexes. RNA and proteins can be captured at a fast acquisition rate. This work used 1000 or 5000 fps, which is fast enough to capture RNA and protein movement in the nucleus. In addition, RNA can form granules and participate in chromosome compaction. How RNA and nuclear proteins contribute to the signals in this work has not been discussed or calibrated. The diffusion constants of chromatin measured by iSCORS are much faster than those reported by other groups which can probably be explained by contributions from freely diffusing particles. The authors should address this issue to be considered by the journal.

We thank the reviewer for pointing out the concerns about the specificity of the DLS signal to chromatin. Following the suggestions, we have examined the DLS signal of nuclear speckles, one of the large protein-RNA complexes in the nucleus. Under the guidance of fluorescence markers, we found that the nuclear speckles generate a much weaker DLS signal than chromatin. We believe this is because protein-RNA complexes lack high-ordered structures like chromatin, and thus our measured scattering signal is dominated by the dense and bulky molecular complexes of chromatin.

Although the iSCORS maps are highly correlated with the H2B fluorescence image, there are minor discrepancies, indicating that non-chromatin structures also generate measurable DLS signals. As the primary aim of this study is to explore chromatin condensation dynamics by label-free iSCORS imaging, we intend to avoid fluorophore labeling and thus do not have markers to ensure absolutely signal specificity to chromatin. Thus, in this study, we focus on the global chromatin condensation that is determined by calculating the spatial median value of iSCORS data over the cell nucleus, without interpreting the subnuclear structures and dynamics.

Further justifications and supporting data are presented in the responses to specific comments.

The second major concern is the conclusion of chromatin condensation during transcription inhibition by DRB. It is known that inhibition of pol II activity induced redistribution of nucleolar proteins around the nucleoli. Such a phenomenon is driven by RNA and nucleolar proteins, not chromatin. The figures shown in this manuscript look like the reorganization of nucleoli, which could serve as evidence that iSCORS does not distinguish chromatin from RNA-protein complexes. The authors should provide data from parallel approaches with direct evidence to strengthen the conclusions.

We acknowledge that the transcription inhibitors act on the pol II without direct effect on chromatin configurations. However, we note that the chromatin organization is closely connected to the transcription activities, as revealed in previous studies. For example, active RNA polymerase is found essential for producing and maintaining decondensed chromatin (DOI: 10.1083/jcb.200011069; DOI: 10.1083/jcb.200809196). Additionally, chromatin compaction induced by transcription inhibitors has been measured by fluorescence-based techniques (FRET/FLIM and DNA stains) at the relevant spatiotemporal scales to our study (DOI: 10.1371/journal.pone.0067689; DOI: 10.1016/j.molcel.2021.06.009).

In this revision, we employed deviation-based image analysis and were able to reproduce the detection of chromatin compaction caused by transcription inhibitors using our fluorescence image data of the DNA stain of Hoechst. All these indicate that chromatin condenses when transcription is interrupted. Finally, although the iSCORS signal is not perfectly specific to chromatin, we confirm that the iSCORS signal is dominated by chromatin and thus our current data analysis represents the global chromatin compaction changes upon transcription inhibition.

The heterogeneity of chromatin organization in the nucleus has been shown by other groups. The authors should include more references to clarify the existing knowledge and the significance of their discovery.

Indeed, several studies have reported cellular heterogeneity in chromatin states using sequence-based methods (DOI: doi: 10.1038/nature14590; DOI: 10.1038/s41587-022-01250-0; DOI: 10.1038/s41576-019-0114-6). These techniques provide opportunities to reveal cellular states under various physiological or pathological conditions, allowing for the detection of unknown or rare cell types, and unraveling cell type-specific differences and dynamics. Moreover, heterogeneity in chromatin compaction, observed in stem cells via fluorescence microscopy, has been found to reflect their differentiation status (DOI: 10.7554/eLife.83444).

In our study, we examine chromatin compaction in a stable U2OS cell line, which is considered a homogeneous cell sample with minimal cellular heterogeneity. The considerable differences in chromatin condensation levels among individual cells, as revealed in our data, are surprising. This underscores the inherent cellular heterogeneity in chromatin organization, a finding that has not been available until now.

Page 9: “These techniques provide opportunities to reveal cellular states under various physiological or pathological conditions, allowing for the detection of unknown or rare cell types, and unraveling cell type-specific differences and dynamics. Moreover, heterogeneity in chromatin compaction, observed in stem cells via fluorescence microscopy, has been found to reflect their differentiation status.⁵⁴ In our study, we examine chromatin compaction in a stable U2OS cell line, which is considered a homogeneous cell sample with minimal cellular heterogeneity. The considerable differences in chromatin condensation levels among individual cells, as revealed in our iSCORS data, are unexpected. This underscores the inherent cellular heterogeneity in chromatin organization, a finding that has not been available until now.”

Other than the above concerns, suggestions with more details are listed below:

Comments:

1. The major concern for this manuscript is that the nucleus is composed of many molecules – DNA, RNA, protein, and metabolites. Although the authors showed a correlation between the VDLS map and the H2B-mCherry image, inconsistent regions can still be found, such as certain bright spots in the H2B-mCherry aligned with dark regions in the VDLS map and vice versa. An interesting question is whether the inconsistent regions reflect nucleosome-free DNA (open chromatin) or RNA condensates/clusters. I recommend the authors provide correlation data of their VDLS map with a live-cell DNA stain, such as Vybrant DyeCycle Ruby Stain, which labels the entire genome in a live setting.

As our reply to the general comments above, iSCORS can measure the DLS signal from RNA-protein complexes, albeit these signals are notably weaker compared to those created by chromatin. This phenomenon is highlighted by the diminished iSCORS signal from nucleoli (as displayed in the figure below where the nucleoli are marked with fluorescence dye SYTO RNASelect), compared to chromatin. Besides nucleoli, we also examine the iSCORS signal of nuclear speckles—dynamic RNA-protein composite structures situated in the interchromatin zones of the cell nucleus—identified via immunofluorescence labeling against the SON nuclear protein. The results affirm that nuclear speckles generate a weaker iSCORS signal from chromatin (figure below), reinforcing the premise that chromatin is the primary contributor to the iSCORS signal.

Using fluorescence labeling against nucleoli and nuclear speckles, we examine the *i*SCORS signal intensity created by nucleoli and nuclear speckles, the major RNA-protein complexes in cell nuclei. The live cell *i*SCORS imaging was performed on cells cultured on coverglass with grid markers. Then, the cells are chemically fixed and stained. Nucleoli are labeled by SYTO RNASelect, and nuclear speckles are marked by immunostaining against SON nuclear protein. The same cells are identified on a confocal microscope for fluorescence imaging. (a) V_{DLS} maps of representative cells, and corresponding confocal fluorescence images of chromatin (H2B-mCherry, red) and nucleoli (SYTO RNASelect, green). The locations of nucleoli are indicated by

the blue arrows, exhibiting low chromatin density and low V_{DLS} intensity. (b) V_{DLS} maps of representative cells, and corresponding confocal fluorescence images of chromatin (H2B-mCherry, red) and nuclear speckles (immunofluorescence against SON, green). The locations of nuclear speckles are indicated by the blue arrows, exhibiting low chromatin density and low V_{DLS} intensity.

Based on the observation that RNA-protein complexes generate weak iSCORS signals, we expect nucleosome-free DNA, or open chromatin, to yield a weak signal due to its low molecular mass. Following the reviewer's suggestion, we verify this hypothesis by examining the similarity between the iSCORS V_{DLS} map and DNA stain of Hoechst, in comparison to that of the iSCORS V_{DLS} map and H2B-mCherry (Figure below). The three images (V_{DLS} , H2B, Hoechst) exhibit high correlations, consistent with our argument that the iSCORS signal is mainly generated by chromatin. However, occasional discrepancies between the V_{DLS} and fluorescence images are noted, which are believed to stem from unrelated scattering signals from non-chromatin structures, such as nuclear bodies and nuclear scaffolds. Given that the linear scattering signal is ubiquitous, achieving absolute specificity to chromatin is challenging. Therefore, this study refrains from interpreting the subnuclear structures depicted in iSCORS maps. Our conclusions are exclusively drawn from the analysis of **global chromatin condensation states**, determined by calculating the spatial median value of the image data.

Representative V_{DLS} map and corresponding confocal fluorescence images of chromatin (H2B-mCherry) and DNA stain (Hoechst 33342), showing high correlations. The fluorescence images of H2B-mCherry and Hoechst exhibit very minor variations, potentially attributable to nucleosome-free DNA and the selective binding affinity of DNA stains (Hoechst predominantly binds to A-T rich region; DOI: 10.3390/chemosensors6020018).

Page 5: “With the guidance of fluorescence labels, we confirm a high correlation between the V_{DLS} map and the confocal fluorescence chromatin image (H2B-mCherry), both of which resolve numerous chromatin-depleted regions that correspond to nucleoli and RNA-filled compartments (Figure 1f and Figure 1g, more image data in Supplementary Fig. 6).³⁷ Meanwhile, we measure much weaker DLS signals from RNA-protein complexes, such as nucleoli (labeled by SYTO RNASelect) and nuclear speckles (immunostained against the nuclear protein SON), compared to those from chromatin (Figure 1f and Figure 1g, more in Supplementary Fig. 6). Figure 1d displays the quantitative comparisons of the V_{DLS} signals created by chromatin, nuclear speckles, and nucleoli, showing that chromatin contributes the strongest signal. We believe this is because protein-RNA complexes lack high-ordered structures like chromatin, and thus the measured DLS signal is dominated by the large and dense molecular complexes of chromatin.”

Page 6: “For quantitative analysis, we calculate the spatial median values within the nuclear region to represent the global chromatin configuration, mitigating bias from nonspecific and spatially localized nuclear structures such as nucleoli and other RNA-protein complexes.”

Page 12: “Despite the nucleus being composed of many non-chromatin nuclear proteins and RNA, iSCORS signals predominantly originate from chromatin, evidenced by the strong correlation between iSCORS maps and fluorescence chromatin images. We attribute this to the enhanced scattering intensity of chromatin due to its dense, high-ordered structures, a feature that RNA-protein complexes lack. Occasional discrepancies between the iSCORS and fluorescence chromatin images are noted, which suggest unrelated scattering signals from non-chromatin structures, such as nuclear bodies and nuclear scaffolds. Given that the linear scattering signal is ubiquitous, achieving iSCORS signal specificity exclusively for chromatin remains a complex task. Instead, this study refrains from interpreting the subnuclear structures depicted in iSCORS maps. Our conclusions are drawn from the analysis of global chromatin condensation states, determined by calculating the spatial median intensity across the images.”

2. The reported apparent diffusion coefficient ($0.92 \pm 0.07 \mu\text{m}^2/\text{s}$, line 164) is approximately 10 times larger than the reported mRNA diffusion constants in the nucleus (PMID: 12546792). Given that RNA is less constrained than chromatin, an explanation for this in the discussion would help the readers understand the meaning of these parameters.

Chromatin displays subdiffusive behavior within the live cell nuclei, leading to a decrease in the apparent diffusion coefficient (D^*) as the measurement timescale extends (DOI: 10.1091/mbc.E14-06-1127; DOI:10.1126/sciadv.abn5626; DOI:10.1073/pnas.1907342116). When using iSCORS at a timescale of 1 ms, a D^* of about $1 \mu\text{m}^2/\text{s}$ is measured. However, extending the iSCORS measurement timescale to 5 ms through downsampling causes D^* to reduce by five folds (illustrated in the figure below). This phenomenon accounts for the lower D^* values reported in fluorescence-based single loci tracking studies, which typically capture data at a slower speed (around 10-100 ms timescale). Aligning with this observation, fluorescence recovery after photobleaching (FRAP) methods, which operate over even longer timescales (~500 ms), have recorded substantially lower D^* values, on the order of $0.01 \mu\text{m}^2/\text{s}$ (DOI: 10.1186/s13072-016-0093-1). On the other hand, the D^* we measured is approximately an order

of magnitude slower than the diffusion of free eGFP in cell nuclei, supporting that iSCORS measures the diffusion of chromatin fibers, not freely diffusing nuclear proteins.

Page 5:

“An apparent diffusion coefficient of $0.92 \pm 0.07 \mu\text{m}^2/\text{s}$ is measured for chromatin in the millisecond timescale. Such D^* is approximately an order of magnitude slower than the diffusion of an inert protein in the cell nucleus, suggesting that iSCORS measures the diffusion of chromatin fibers instead of freely diffusing nuclear proteins.³⁴ Furthermore, iSCORS measurement is in a millisecond timescale, capturing local molecular diffusion events that are faster than the chromatin movements measured by fluorescence-based methods, such as tracking and bleaching experiments that typically work over a much longer timescale of tens to hundreds of milliseconds.³⁵ This is due to the subdiffusive motion of chromatin, leading to a decreasing D^* when the timescale increases.^{14,15,36} We confirm this phenomenon by increasing the measurement timescale of iSCORS to 5 ms, and indeed a significantly reduced D^* is measured, consistent with the behavior of chromatin subdiffusion (Supplementary Fig. 5).”

3. How the effective concentration of each treatment was determined is not clear. Such information is important because some reagents showed additional functions (e.g., triggering apoptosis) when used in a high concentration. Please include this information in the manuscript. We followed the conditions of drug treatments reported in previous studies that effectively modify the chromatin compaction or inhibit the gene transcription without introducing noticeable apoptosis. When applicable, we intend to shorten the drug treatment duration to further reduce the long-term side effects on the cells. Our drug treatment conditions are listed below. These respective references are added in the Methods section.

	Our treatment conditions	Previous reports	Refs
NaB	5 mM, 2 hr	2.5mM, 24hr	DOI: 10.1093/nar/gkz373
TSA	500 nM, 2 hr	500 nM, 24hr	DOI: 10.1242/jcs.01293
ATP depletion	10 mM NaN ₃ + 50 mM 2-deoxy-glucose, 30 minutes	10 mM NaN ₃ + 50 mM 2-deoxy-glucose, 10 minutes	DOI: 10.1371/journal.pone.0067689
ActD	1 μM , 2 hr	<5 μM , <2 hr	DOI:

			10.1073/pnas.1814965116 PMID: 25932119
DRB	200 μ M, 2 hr	100 μ M, 2 hr	DOI: 10.1016/j.molcel.2017.06.018
alpha-AM	100 μ M, 2 hr	~100 μ M, 2 hr	DOI: 10.1083/jcb.201811090

Page 22: “We followed the conditions of drug treatments reported in previous studies that effectively modify the chromatin compaction or inhibit the gene transcription without introducing noticeable apoptosis. When applicable, we intend to shorten the drug treatment duration to further reduce the long-term side effects on the cells.”

4. Treatment of DRB does not have direct effects on chromatin states. DRB inhibits transcription by inhibiting RNA elongation of RNA pol II, which stops the synthesis of new RNA molecules. However, DRB neither degrades the existing RNA nor alters chromatin states (considering chromatin compaction is primarily regulated by histone modifications and SMC complexes.) The reference cited (#41) confirmed that treatment of DRB alone has no obvious change to chromatin compaction. The chromatin condensation observed by the authors (line 277) during DRB treatment is mostly likely changes in RNA or protein, not DNA. The authors should simultaneously use a live-cell DNA stain (e.g., Vybrant DyeCycle Ruby Stain) and a live-cell RNA stain (e.g., SYTO) to strengthen their conclusions and explain the rationale of their results.

Although DRB inhibits the transcription by interrupting the functional activity of RNA polymerase II, which does not directly modify the chromatin configuration, previous studies reported chromatin compaction induced by DRB based on fluorescence-based FLIM-FRET (DOI: 10.1371/journal.pone.0067689, acknowledged as Ref. 41 in our manuscript) and DNA stains (DOI: 10.1016/j.molcel.2021.06.009). Specifically, in this early study by FLIM-FRET, two fluorescence proteins, EGFP and mCherry, are randomly fused to histone protein H2B. To probe the chromatin compaction, the fluorescence lifetime of EGFP is measured, as it is sensitive to its relative proximity to mCherry due to the FRET effect. It was observed that DRB treatment leads to a mild reduction in EGFP fluorescence lifetime across cell nuclei, indicating a decrease in nucleosome proximity, a signature of chromatin condensation. In addition, it was found that the DRB-induced condensation is less significant than that caused by ATP depletion, which agrees with our iSCORS data in this manuscript.

Besides the FRET-FLIM experiments, using fluorescence DNA staining and deviation-based image analysis, previous studies also observe chromatin compaction under various drug treatments that inhibit gene transcription activities of RNA polymerase II, including DRB, ActD, and α -Amanitin (DOI: 10.1016/j.molcel.2021.06.009). Moreover, active RNA polymerase is found essential for producing and maintaining decondensed chromatin (DOI: 10.1083/jcb.200011069; DOI: 10.1083/jcb.200809196).

All these early studies show that the DRB treatment induces global chromatin compaction, although the underlying mechanisms remain unclear.

In this revision, we have verified the compaction of chromatin resulting from transcription inhibition in our lab through the examination of fluorescence confocal images of DNA stains. These fluorescence images reveal that treating cells with transcription inhibitors results in spatially heterogeneous fluorescence intensities, a signature of chromatin condensation (see the image data below). Detecting this alteration is difficult for CCP analysis as reported in our original manuscript. Following the comment of Reviewer 3, we employ a different image analysis to the fluorescence data of DNA Hoechst stain, which successfully reveals the mild chromatin condensation induced by the transcription inhibitors (data below). This new analysis calculates 'the coefficient of variation (CV)', a value that represents the spreading of fluorescence intensity in an image, which proves more effective in assessing chromatin condensation levels compared to edge detection-based CCP (DOI: 10.1016/j.xpro.2021.100865). In the revised manuscript, we replaced the original CCP data with the new CV results.

Chromatin compaction upon transcription inhibition detected by confocal fluorescence imaging of Hoechst. Left: confocal fluorescence images of representative cells stained by Hoechst under treatments of transcription inhibition compared to the control. Right: Coefficient of variation (CV) calculated based on the Hoechst confocal fluorescence image for different treatments. An increase in the CV was measured, indicating an increase in chromatin condensation level.

We examine the Vybrant DyeCycle Ruby DNA stain as suggested by the reviewer. We found that the images of Vybrant DyeCycle Ruby DNA stain are very similar to those of Hoechst, including the CV values under treatments (see the images and data below). However, significant localizations of Vybrant DyeCycle Ruby DNA stain were found in the cytoplasm. Therefore, we choose to use Hoechst as our DNA stain for this study.

Comparison of Hoechst 33342 and Vybrant DyeCycle Ruby for DNA staining in cell nuclei, using fluorescence confocal microscopy for imaging. Alongside normal cells, two treatments modified global chromatin compaction: ATP depletion, inducing condensation, and NaB, inducing decondensation. The fluorescence imaging results of both dyes appeared visually similar across all conditions. The coefficient of variation (CV) from the imaging reflected chromatin compaction changes due to treatments, indicating that both Hoechst 33342 and Vybrant DyeCycle Ruby dyes perform comparably in DNA staining and revealing chromatin compaction states. Scale bar: 5 μ m.

Finally, we verify whether the iSCORS signal change under the DRB treatment is associated with chromatin condensation by examining the correlation between the iSCORS condensation map and fluorescence H2B image. We confirm a high correlation between the iSCORS condensation map and the H2B image (Figure below), indicating that the iSCORS signal remains highly chromatin dominant under the DRB treatment. Based on the above examinations, we conclude that iSCORS measures the chromatin condensation process induced by DRB.

Page 8: “Our observation of chromatin compaction upon blocking the transcription activity of RNA Pol II is consistent with previous works. While these transcription inhibitors act on the RNA pol II without direct effect on chromatin configurations, early studies have revealed a close connection between the chromatin organization and transcription activities. For example, active RNA polymerase is found essential for producing and maintaining decondensed chromatin.⁴⁸ Additionally, chromatin compaction induced by transcription inhibitors has been measured by fluorescence-based techniques.^{49,50} The enhanced chromatin condensation in our experiments is further supported based on an analysis of fluorescence intensity distribution of DNA stain of the cell nuclei (Figure 4c and Supplementary Fig. 13).³⁸”

5. In line 293, the authors claimed that “The observed chromatin condensation dynamics induced by DRB treatment, depicted in Fig 4g, providing clear evidence of chromatin condensation...” This statement needs additional evidence from DNA-specific experiments.

Please see our response to comment #4 where the image data of fluorescence DNA stains indicate chromatin condensation.

6. The authors mentioned the results that chromatin in mammalian cells undergoes anomalous subdiffusion. In addition to single nucleosome tracking (ref [14, 15] proving chromatin dynamics averaged across the whole nucleus, similar results were also reported by other approaches, such as LaO and CRISPR-based imaging that can probe the dynamics of specific genomic loci.

We thank the reviewer for pointing out these relevant studies showing subdiffusion of gene loci. We have added two early works in the reference (listed below).

CRISPR-based labeling and tracking of single genomic loci reveals subdiffusion of single (DOI: 10.1101/gr.260018.119; 10.1038/s41598-021-91787-y)

Page 5: “Early studies show that chromatin in mammalian cells undergoes anomalous subdiffusion, measured by tracking single nucleosome and specific genomic loci.^{14,15,31,32}”

7. The authors propose two quantities (VDLS, D^*) to characterize chromatin condensate dynamics. The log-log plot of these two quantities shows a line with slope 3 followed by the dependence of VDLS $\sim (1/D^*)^3$. What is the meaning of the y-intercept on the log-log plot? Is it a cell-type-dependent constant or a chromatin-location-dependent constant? The authors should discuss it.

The y-intercept indicates the macroscopic molecular density of chromatin, more specifically, the average moving mass density within the optical detection volume (approximately $0.5 \times 0.5 \times 1 \text{ micron}^3$). We emphasize that this macroscopic chromatin density does not correspond to the level of chromatin condensation, which is a measure of chromatin compaction on the nanoscale. For instance, it is possible to observe a consistent moving mass density at different levels of condensation. To demonstrate this, we have shown that the overall mass density of a nanoparticle colloid can be determined using the y-intercept on the $V_{DLS}-1/D^*$ graph (Supplementary Fig. 4), and this method is effective for particles of diverse sizes where different particle sizes represent different condensation of mass at the nanoscale.

To investigate whether the y-intercept varies based on location in a cell nucleus, we charted the iSCORS data for each pixel inside a cell nucleus on a V_{DLS} - $1/D^*$ graph (panel a below), showing a spread of data points that reflect the distribution of molecular density and condensation level. We plotted a map of y-intercept values for a representative cell nucleus (panel b), which shows a high correlation with the confocal fluorescence image of chromatin (H2B-mCherry, panel c). It not only supports the fact that our iSCORS signal is dominated by the scattering of chromatin but also validates our interpretation that the y-intercept represents chromatin density.

Minor discrepancies can be found in the y-intercept map and the H2B-mCherry fluorescence image. As stated in the response to the general remarks, in this study, we focus on the global chromatin configurations without interpreting subnuclear structures and dynamics. By calculating the spatial median value over the whole nucleus, we measure the mobility, density, and compaction of the chromatin-rich region of the nucleoplasm.

Finally, we perform iSCORS measurements on two additional cell types: Chinese Hamster Ovary (CHO) cells and Human foreskin BJ fibroblasts. Our findings reveal that their chromatin densities closely resemble those of human osteosarcoma U2OS cells (data below, their differences are $\sim 10\%$). This similarity suggests a conservation of chromatin density across mammalian cells, indicating potential similarities in the underlying mechanisms governing chromatin organization and dynamics. Further investigations are needed to explore potential differences in chromatin configurations in other cell types.

Page 7: "It is informative to examine the meaning of the y-intercept value of the line with a slope of three in the log-log plot of V_{DLS} against $1/D^*$. From the iSCORS model and nanoparticle data

(Supplementary Fig. 8), the y-intersect value indicates the overall mass density. We validate the interpretation of the y-intersect value as mass density by mapping the y-intersect value of a cell nucleus onto a spatial representation. The derived y-intersect map shows a high correlation with the chromatin fluorescence image, supporting our model that the y-intersect denotes mass density (Supplementary Fig. 9).”

“Interestingly, the mass densities of different mammalian cell lines also exhibit a similar constancy, with a variation of less ~10% (Supplementary Fig. 11).”

8. Related to the comment above, in Fig 2b, the authors performed the perturbation of global chromatin condensation by chemicals. These data seem not best fitted into the same line with a slope 3 but into parallel lines with the same slope 3 but different y-intercepts. The authors are encouraged to discuss the meaning of these discrepancies.

We attribute the spreading of the data points with respect to a single line with a slope of 3 in a $V_{DLS}-1/D^*$ plot to cell heterogeneity in chromatin density and our measurement errors. As illustrated by the data of more than 100 nuclei in a normal state (figure below), ~15% variations are observed between cells, which underscores our overall measurement uncertainty. This also explains the data spreading in Figure 2b for the data of different global chromatin condensations induced by chemicals.

Page 7: “The chromatin mass density of individual cells is consistent, within a variation of ~15% (Supplementary Fig. 10), set by the inherent cell heterogeneity and measurement uncertainty.”

Supplementary Fig. 10

9. In Figure 2c, the authors should explain how the chromatin condensation levels were calculated. The authors indicated that chromatin condensation levels were generated by the projection of data points on the line of slope 3. Are the data points projected on the same line? After projection, how to read off the values (for chromatin condensation level)? The authors should explain why this value can be used to characterize the level of chromatin condensation.

The level of chromatin condensation in a cell is defined as the distance between the cell's iSCORS data point and the noise baseline data point, after projection onto a line in the log-log plot of V_{DLS} and $1/D^*$. It is important to note that the calculated condensation level does not depend on the choice of the y-intercept. In this case, the noise baselines for V_{DLS} and $1/D^*$ are set at 10^{-5} and 10^{-3} , respectively.

This definition is derived from nanoparticle data (as shown in response to comment #7) and is further validated by the results from cell experiments with drug treatments that alter the global chromatin condensation (see Figure 2b). The data shown in Figure 2c correspond to the measured chromatin condensation levels normalized by that of normal cells. The fundamental concept of this derivation posits that chromatin condensation increases the size of chromatin nanodomains. This increase enhances the scattering intensity and reduces the mobility. In analyses of apparent free diffusion, changes in chromatin condensation—with a constant overall mass density—cause the cell data point to move along a line with a slope of three.

A new section that described the quantification of chromatin condensation level is added in Methods:

Page 22:

“Quantification of chromatin condensation level by iSCORS

The level of chromatin condensation is defined by the distance between the iSCORS data point and the noise baseline data point, projected onto a line in the log-log plot of V_{DLS} and $1/D^*$. For this assessment, the noise baselines for V_{DLS} and $1/D^*$ are set at 10^{-5} and 10^{-3} , respectively. The calculated condensation level does not depend on the choice of the y-intercept value of the line being projected.”

10. The authors observed similar distributions of chromatin condensation during the interphase. The authors should discuss the findings using other approaches, such as global chromatin domain analysis and individual chromatin compactions.

Reviewer 1 raises a similar question (comment #9) about our findings of similar chromatin compaction for interphase cells. We iterate our reply here:

The prevailing view is that chromatin structure undergoes significant changes throughout the interphase, reflecting distinct nuclear activities and chromatin remodeling needs characteristic of each phase. However, we note that changes in chromatin architecture across different cell phases greatly depend on the spatial and temporal scales of the measurements. For example, in the timescale of approximately 10 seconds, single telomeres exhibit different diffusion characteristics in the interphases where chromatin diffusion is more restricted in the S and G2 phases than in the G1 phase (DOI: 10.1016/j.isci.2022.104197). On the other hand, in a shorter timescale of sub-second, telomere diffusion exhibits no significant differences throughout the interphase. Another earlier study by single nucleosome tracking also reported that local chromatin motion in ~50 ms timescale is similar throughout the interphase (DOI: 10.1016/j.molcel.2017.06.018). These

observations match with our iSCORS data, which measures chromatin fluctuations at the small spatial and temporal scales.

We emphasize that our study only shows that the global chromatin configuration does not change significantly during the interphase. We did not explore the local and transient chromatin reorganization which may have a closer association with specific nuclear events in the interphase, e.g., DNA replication in the S phase. Such exploration is complicated by cell movement, posing challenges to monitoring chromatin dynamics at specific chromosomes/chromosomal segments without fluorescence guidance.

Page 10: “It is generally believed that the organization of chromatin changes considerably during interphase, reflecting the unique nuclear activities characteristic of each phase and their associated chromatin remodeling requirements. We point out that the variations in chromatin architecture and dynamics across different cell phases are highly dependent on the spatial and temporal scales of the measurements. For example, in the timescale of approximately 10 seconds, single telomeres exhibit different diffusion characteristics in the interphases where chromatin diffusion is more restricted in the S and G2 phases than in the G1 phase.⁵⁶ On the other hand, in a shorter timescale of sub-second, telomere diffusion exhibits no significant differences throughout the interphase. Another earlier study by single nucleosome tracking also reported that local chromatin motion in ~50 ms timescale is similar throughout the interphase.⁹ iSCORS characterizes chromatin architectures in the short spatiotemporal scales and our data agree with these previous observations that chromatin dynamics are nearly constant throughout the interphase.”

Minor comments:

11. Small grammar issues, such as a missing period in line 87, should be fixed.

This mistake has been corrected in the revised manuscript.

12. Reference 8 has duplicated “doi:.” One of them should be removed.

This mistake has been corrected in the revised manuscript.

13. The nucleus boundary in Figure 1d and 1e does not fit the nuclear boundary in 1f. Please provide the information regarding how the nuclear boundary was determined.

The nuclear boundary was identified in the C map using a machine-learning model from our previous study (ref. 25), as described in the Methods section. For clarification, we add a statement of nuclear segmentation in the figure legend.

14. The source of fluorescence in the fluorescence images in Figure 2a is unclear. Please provide the information in the figure legend.

We have added the information on confocal fluorescence images of chromatin (H2B-mCherry).

15. Two graphs are labeled as Figure 5c in the legend.

We apologize for the mistake. It has been corrected.

16. Please include the number of cells in Figure 5d.

We have added the number of cells in the histograms in Figure 5d: N=37 for G1, N=104 for S, and N=43 for G2.

17. The information regarding the source of the U2OS H2B-mCherry cells is missing.

The U2OS cells are obtained from ATCC. A stable cell line U2OS H2B-mCherry was established by overexpressing H2B-mCherry (pH2B_mCherry_IRES_neo3, Addgene, #21044), established by a local company (Omisc Bio, Taiwan). This information is added in Methods.

Reviewer #3 (Remarks to the Author):

In this manuscript, Hsiao et al. have developed a label-free bright-field microscopic method, termed iSCORS, to investigate chromatin condensation in living cells. This method extends the previously reported interferometric scattering microscopy technique. The spatial resolution is a similar range to the spinning-disk confocal microscopy. Chromosome condensation levels were assessed by obtaining the temporal variance of dynamic light scattering (VDLS), the apparent diffusion coefficient (D^*), and the interference contrast (C). Control experiments using drugs known to trigger chromosome condensation (actinomycin D, 2-deoxyglucose, and sodium azide) and decondensation (trichostatin A and sodium butyrate) demonstrated that VDLS, $1/D^*$, and C could measure chromosome condensation levels. Upon treatments with transcription inhibitors, chromatin condensation levels were increased. The condensation induced by DRB was reversed upon its removal. During the cell cycle, chromosome condensation levels were relatively constant during the G1 and S phases, while being slightly decreased during the G2.

The technique is unique and potentially powerful for understanding chromatin structure at the nanoscale level in living cells without fluorescence labels. However, the results provide little biological advance. The label-free technique should offer advantages such as high-speed and less-toxic imaging at high resolution, but none of these were explored. I believe it is essential to demonstrate findings that are exclusively obtainable using this new technique for publication in a high journal.

Our research presents iSCORS as an innovative method for assessing chromatin dynamics in living cells without the need for labels. To our knowledge, our study achieves unprecedented accuracy in depicting chromatin diffusion and compaction in a label-free manner. While our findings are mostly consistent with previous fluorescence-based studies, the iSCORS imaging strengthens the evidence by eliminating potential labeling artifacts, thereby providing a truer reflection of chromatin behavior within living cells.

Moreover, iSCORS enables precise quantification of chromatin condensation in individual cells, free from the variability introduced by inconsistent fluorescence labeling. Through iSCORS, we have uncovered a surprising degree of heterogeneity in chromatin compaction within a stable cell line. Additionally, by taking advantage of long-term imaging capabilities, we have found, for the first time, the spontaneous fluctuations in chromatin compaction over 15 hours. These insights attest to the novelty and significance of our research.

Specific points

1. Although there are some intriguing data that may contradict previous findings, an orthogonal approach is essential for validating the conclusion. It has been shown that DRB treatments do not induce chromatin condensation and rather induce decondensation (e.g., doi.org/10.1006/excr.1996.0124).

The paper mentioned by the reviewer reports the reorganization of satellite DNA and nucleolar necklaces upon the treatment of DRB. We note that these structures are specific to repetitive

DNA sequences or nucleolar fibrillar substructures, which may not necessarily represent the chromatin remodeling over the whole genome as what iSCORS measured in our study.

On the other hand, using the fluorescence-based technique of FRET/FLIM, prior studies report mild global chromatin condensation induced by DRB, which is consistent with our observation (DOI: 10.1371/journal.pone.0067689, acknowledged as Ref. 49 in our revised manuscript). Briefly, in this early study, two fluorescence proteins, EGFP and mCherry, are randomly fused to histone protein H2B. To probe the chromatin compaction, the fluorescence lifetime of EGFP is measured, as it is sensitive to its relative proximity to mCherry due to the FRET effect. It was observed that DRB treatment leads to a reduction in EGFP fluorescence lifetime across cell nuclei, indicating a decrease in nucleosome proximity, a signature of chromatin condensation. In addition, it was found that the DRB-induced condensation is less significant than that caused by ATP depletion, which agrees with our iSCORS data in this manuscript. Besides the FRET/FLIM experiments, using fluorescence DNA staining and deviation-based image analysis, previous studies also observe chromatin compaction under various drug treatments that inhibit gene transcription activities of RNA polymerase II, including DRB, ActD, and α -Amanitin (DOI: 10.1016/j.molcel.2021.06.009). All these indicate a rise in DNA compaction upon transcription inhibition, consistent with our findings.

Page 8: “Our observation of chromatin compaction upon blocking the transcription activity of RNA Pol II is consistent with previous works. While these transcription inhibitors act on the RNA pol II without direct effect on chromatin configurations, early studies have revealed a close connection between the chromatin organization and transcription activities. For example, active RNA polymerase is found essential for producing and maintaining decondensed chromatin.⁴⁸ Additionally, chromatin compaction induced by transcription inhibitors has been measured by fluorescence-based techniques.^{49,50} “

In addition, chromatin condensation induced by actinomycin D has been detected in many studies using GFP-tagged histones and Hoechst staining, unlike stated in Introduction and shown by the authors. The analytical method involving edge detection used in the manuscript may not be suitable to assess the chromatin compaction. Consider using another method, such as the standard deviation-based measurements.

We appreciate very much the reviewer for suggesting the deviation-based image analysis on fluorescence DNA images. We find that the ‘coefficient of variation (CV)’, a value that represents the spreading of fluorescence intensity in an image, is effective in assessing chromatin condensation levels. Specifically, for a single nucleus, a CV value is defined as σ/μ , where σ and μ denote the standard deviation and the mean value of fluorescence intensity within the nucleus, respectively. This approach proves more sensitive to the chromatin compaction level within the entire nucleus compared to edge detection-based CCP.

By following the protocol and the algorithm reported in DOI: 10.1016/j.xpro.2021.100865, we successfully detect chromatin compaction due to transcription inhibitors, e.g., DRB and ActD. In the revised manuscript, we replaced the original CCP data with the new CV results (Figure 4c and Supplementary Fig. 7). We also revised the statement on DNA stains Introduction accordingly.

Quantitation of chromatin condensation of single live cell nuclei under different drug treatments by coefficient of variation (CV) analysis. (a) Confocal fluorescence images of cell nuclei stained by Hoechst under different drug treatments that modify chromatin compaction states. (b) The coefficient of variation (CV) calculated from the Hoechst fluorescence confocal images under different drug treatments. Higher CVs are measured for chromatin compaction (2-DG+NaN₃ and ActD), and lower CVs are measured for chromatin decondensation (NaB and TSA).

Figure S13 Hoechst fluorescence confocal images of cell nuclei under transcription inhibition.

Page 6: “The effectiveness of these treatments on modifying chromatin condensation states is confirmed by analyzing the fluorescence intensity distribution of DNA stains based on coefficient of variation (CV), a method previously established to estimate chromatin compaction (Method and Supplementary Fig. 7).³⁷”

Page 8: “The enhanced chromatin condensation in our experiments is further supported based on an analysis of fluorescence intensity distribution of DNA stain of the cell nuclei (Figure 4c and Supplementary Fig. 13).³⁷”

2. Although the iSCORS technique has a high spatial resolution, the analyses are all the average of a single nucleus and the outcome is not very novel. Since local chromatin condensation levels change during DNA replication, the technique could be used to demonstrate such changes during the S phase. It would be interesting if the temporal order of replication from euchromatic to heterochromatic regions is detected by iSCORS.

We thank the reviewer for recommending experiments with high spatiotemporal resolution to highlight the potential benefits of iSCORS. Our present imaging systems are indeed capable of capturing iSCORS data at high spatial and temporal resolutions. Although we are keen to investigate the transient and localized chromatin remodeling capabilities of iSCORS, it is imperative to be careful about ensuring that the iSCORS signal is specific to chromatin. As we mentioned in response to reviewer 2's comment #1, iSCORS might also detect signals from nuclear bodies and nuclear scaffolds. Consequently, our research currently focuses on the dynamics of global chromatin condensation. In future work, guided by fluorescence markers, we believe localized chromatin remodeling can be studied as the reviewer has suggested.

While this study does not analyze chromatin dynamics with spatial resolution, it is important to highlight that we have uncovered previously unknown aspects of chromatin condensation and dynamics. Our findings include the discovery of cell heterogeneity in chromatin condensation levels during the interphase, irrespective of cell cycle phases, and the identification of spontaneous chromatin condensation dynamics characterized by durations of tens of minutes. These insights attest to the novelty and significance of our research.

Page 10: “We did not explore the local and transient chromatin reorganization which may have a closer association with specific nuclear events in the interphase, e.g., DNA replication in the S phase. Such exploration is complicated by cell movement, posing challenges to monitoring chromatin dynamics at specific chromosomes or chromosomal segments in the current setup.”

REVIEWERS' COMMENTS:

Reviewer #1 (Remarks to the Author):

I believe the authors have thoroughly addressed all the points raised in my previous review. They have carefully analyzed their data and performed additional experiments, which have strengthened their manuscript. I agree that this work is now suitable for publication as it stands.

Reviewer #2 (Remarks to the Author):

The revised manuscript has been improved substantially. The authors have provided additional data to strengthen their conclusions, which clarifies the method and elevates the significance of the work. I have no further questions except for two minor comments:

(1) The authors called U2OS "normal cells" a few times throughout the manuscript (such as on page 6 and in Figure 4 legend). Although I understand they mean "untreated cells", U2OS are osteosarcoma cells that are not considered "normal". The term (normal cells) is usually used to indicate cells without disease. Therefore, "untreated" or its synonyms are better for describing the cells. Cancer cells are also not considered in a normal state. The authors are encouraged to clarify the wording.

(2) On page 8, the authors claimed that DRB, α -AM, and ActD are inhibitors acting on the RNA pol II without direct effect on chromatin configurations. While this statement stands true for DRB and α -AM, it is inaccurate for ActD, which has been shown to bind DNA directly and affect DNA structure (doi: 10.1093/nar/gkt084). The authors are encouraged to rephrase the sentence for better accuracy.

Reviewer #3 (Remarks to the Author):

The revised manuscript has partially addressed the previous concerns and shows significant improvement. However, the biological significance remains weak, and the application of the technique appears limited due to the lack of data at higher spatial or temporal resolutions.

Responses to referees' comments

We thank referees #1 and #3 for endorsing the publication of our revised manuscript. In response to the suggestions of referee #2, we have made several minor adjustments. Specifically, we have replaced the term 'normal cells' with 'untreated cells' throughout the manuscript. In addition, we have removed the inaccurate statement on page 8 regarding the direct action of actinomycin (ActD) on chromatin. We appreciate it very much that referee #2 pointed out this mistake.